# A Neuro-Symbolic Benchmark Suite for Concept Quality and Reasoning Shortcuts

**Samuele Bortolotti**[1*]   **Emanuele Marconato**[2,1*]   **Tommaso Carraro**[3,6]   **Paolo Morettin**[1]
**Emile van Krieken**[4]   **Antonio Vergari**[4]   **Stefano Teso**[5,1]   **Andrea Passerini**[1]

[1] DISI, University of Trento    [2]DI, University of Pisa    [3]Fondazione Bruno Kessler
[4]University of Edinburgh    [5]CIMeC, University of Trento    [6]University of Padova

{name.surname}@unitn.it
tcarraro@fbk.eu
{Emile.van.Krieken, avergari}@ed.ac.uk

## Abstract

The advent of powerful neural classifiers has increased interest in problems that require both learning and reasoning. These problems are critical for understanding important properties of models, such as trustworthiness, generalization, interpretability, and compliance to safety and structural constraints. However, recent research observed that tasks requiring both learning and reasoning on background knowledge often suffer from *reasoning shortcuts* (RSs): predictors can solve the downstream reasoning task without associating the correct concepts to the high-dimensional data. To address this issue, we introduce `rsbench`, a comprehensive benchmark suite designed to systematically evaluate the impact of RSs on models by providing easy access to highly customizable tasks affected by RSs. Furthermore, `rsbench` implements common metrics for evaluating concept quality and introduces novel formal verification procedures for assessing the presence of RSs in learning tasks. Using `rsbench`, we highlight that obtaining high quality concepts in both purely neural and neuro-symbolic models is a far-from-solved problem. `rsbench` is available at: https://unitn-sml.github.io/rsbench.

## 1 Introduction

Although the field of deep learning has made significant progress in developing accurate neural classifiers, end-to-end neural networks struggle with tasks that also require symbolic reasoning on low-level inputs like visual objects [1, 2]. Instead, Neuro-symbolic (NeSy) AI [2–5] promises to improve the trustworthiness of AI systems by integrating perception with symbolic reasoning [6, 7]. This involves extracting high-level *concepts* from the input and reasoning over them with some prior knowledge, *e.g.*, safety constraints, to obtain a prediction. This setup can encourage [8–13] or even ensure [14–16] the output complies with the knowledge.

Recent evidence suggests that, in some problems, NeSy models can achieve high accuracy on the reasoning task by *learning concepts with incorrect semantics*. Such **reasoning shortcuts** (RSs) [17] occur when the knowledge, which acts as a bridge between the given output labels and the concepts [18], allows for inferring the right label using unintended concepts. This can seriously undermine the original purpose of NeSy AI systems, especially in high-stakes scenarios. For instance, in the `BDD-OIA` dataset [19], a model is given a set of traffic laws, and must predict what actions an autonomous vehicle is allowed to perform (*e.g.*, "go" or "stop"). It will believe it obeys these laws by confusing pedestrians for red lights, as both entail the correct action ("stop"). Yet, if – when used

---

*Equal contribution.

38th Conference on Neural Information Processing Systems (NeurIPS 2024) Track on Datasets and Benchmarks.

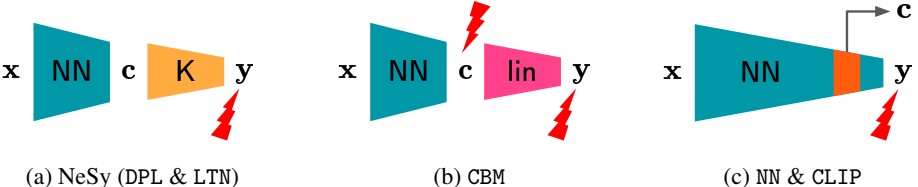

(a) NeSy (DPL & LTN)  (b) CBM  (c) NN & CLIP

Figure 1: **Role of concepts in deep learning models.** (a) NeSy architectures like DeepProbLog (DPL) and Logic Tensor Networks (LTN) map the input $\mathbf{x}$ to concepts $\mathbf{c}$ and reason over these according to prior knowledge to obtain a label $\mathbf{y}$. (b) CBMs are similar, except the prediction is computed by a learned linear layer, making it easy to obtain concept-level explanations of all predictions. (c) Black-box neural networks infer a label $\mathbf{y}$ directly from the input $\mathbf{x}$; concepts $\mathbf{c}$ can be extracted from their latent representation by applying techniques like TCAV [50]. Lighting bolts indicate what variables are usually supervised.

in an out-of-distribution (OOD) task – the vehicle is allowed to cross over red lights in case of an emergency, its preexisting confusion can lead to unfortunate scenarios [20]. RSs impact learnability [18], interpretability of the learned concepts [21–24], and reliability in down-stream tasks [20, 25–28]. At the same time, they can affect most NeSy architectures, regardless of how they are implemented, including approaches based on probabilistic logic [12–14, 16, 29–33], fuzzy logic [8, 34], reasoning in embedding space [35], and abduction [36, 37]. Given their impact, researchers have proposed several mitigation strategies [17, 20, 25, 38–43], yet how to deal with RSs remains an open problem.

Unfortunately, suitable data sets with known RSs are scarce and scattered throughout the literature, hindering research on this challenging problem. Current benchmark suites for learning and reasoning neglect RSs altogether [44] and lack OOD data suitable for investigating their impact, while others are restricted to larger models [45–47]. Simultaneously, available data sets annotated with concept supervision (*e.g.*, CUB200 [48]), which is essential for evaluating concept quality, do not require logical reasoning and do not supply prior knowledge.

**Contributions**. We fill this gap by introducing `rsbench`, an integrated benchmark suite providing all the ingredients needed for systematic evaluation of the impact of RSs and the efficacy of mitigation strategies. `rsbench` comprises: 1) A curated collection of ***tasks*** that require learning and reasoning that are provably affected by RSs. `rsbench` comprises entirely new and already established tasks with different flavors – arithmetical, logical, and high-stakes – along with associated ***data sets*** and ***data generators*** for evaluating OOD scenarios.[2] 2) Python implementations of ***quality metrics*** useful for assessing the impact of RSs on NeSy models and more generally the reliability of concepts learned (explicitly or implicitly) by other concept-based architectures and end-to-end neural networks. 3) a novel algorithm, `countrss`, that exploits automated reasoning techniques [49] to ***verify a priori*** whether a task is affected by RSs and to count them. We showcase `rsbench` by assessing the impact of RSs on the quality of concepts acquired by several deep learning architectures, illustrated in Fig. 1.

## 2 Reasoning Shortcuts: Causes, Consequences, and Scope

We study tasks where models require both learning and reasoning (in short: ***L&R tasks***) to accurately predict a (vector) output $\mathbf{y}$ from low-level inputs $\mathbf{x}$ [20]. First, we assume there is a set of $k$ high-level concepts $\mathbf{c}^*$ associated to the inputs $\mathbf{x}$. Then, we assume the concepts $\mathbf{c}^*$ and prior knowledge K together infer the correct output $\mathbf{y}^*$. The prior knowledge can encode known structural [11] or safety constraints [15] in some formal language (*e.g.*, logical connectives).

**Example 1.** *The* `SDD-OIA` *dataset (detailed in Section 3.3) is a L&R task that contains images $\mathbf{x}$ of 3D traffic scenes, and the goal is to predict one or more allowed actions* {`stop`, `go`, `left`, `right`}. *We assume the correct output depends on binary concepts $c_{\text{grn}}, c_{\text{red}}, c_{\text{ped}}$ encoding whether green lights, red lights, and pedestrians are visible, respectively. The knowledge specifies that if the latter are detected, the vehicle must stop:* K $= (c_{\text{ped}} \lor c_{\text{red}} \Rightarrow y_{\text{stop}})$.

---

[2]In Appendix C.6, we provide a **how-to** guide to the usage of `rsbench`.

Table 1: **List of L&R tasks in `rsbench`.** All columns are described in Section 3.

| | TASK | DATA | | | PROPERTIES | | |
|---|---|---|---|---|---|---|---|
| | | GEN | OOD | CONL | CPLX $\mathbf{x}$ | CPLX K | AMB K |
| ARITHMETIC | MNMath (new) | ✓ | ✓ | ✓ | ✗ | ✓ | ✗ |
| | MNAdd-Half [25] | ✗ | ✓✓ | ✗ | ✗ | ✗ | – |
| | MNAdd-EvenOdd [17] | ✗ | ✓✓ | ✓✓ | ✗ | ✗ | – |
| LOGIC | MNLogic (new) | ✓ | ✓ | ✓ | ✗ | ✓ | ✗ |
| | Kand-Logic [25] | ✓ | ✓ | ✓ | ✗ | ✓ | ✓ |
| | CLE4EVR [17] | ✓ | ✓ | ✓✓ | ✓ | ✗ | ✓ |
| HIGH STAKES | BDD-OIA [20] | ✗ | ✗ | ✗ | ✓ | ✓ | ✓ |
| | SDD-OIA (new) | ✓ | ✓✓ | ✓ | ✓ | ✓ | ✓ |

Reasoning Shortcuts (RSs) have primarily been studied in the context of NeSy models. They usually consist of a **_perception_** module that predicts concepts $\mathbf{c}$ from input $\mathbf{x}$ and a **_reasoning_** module that uses the prior knowledge K to compute an output, as shown in Fig. 1(a). Like most deep learning architectures, NeSy models are trained via maximum likelihood on annotated examples $(\mathbf{x}, \mathbf{y}^*)$, while concept annotation is typically not available. This makes them susceptible to RSs, *i.e.*, they can learn concepts with improper or unclear semantics. For example, in `BDD-OIA` a model can mistake a `pedestrian` for a `red_light` without affecting prediction quality, as both concepts entail the correct `stop` action; more examples can be found in Section 3. Low-quality concepts compromise performance in down-stream decision task [17, 20, 25] – *e.g.*, those that hinge on `pedestrians` and `red_light` being predicted correctly – and in tasks that depend on externally supplied concepts, such as neuro-symbolic formal verification [26, 27], undermining trustworthiness. Moreover, since the concepts' meaning is muddled, concept-based explanations [50, 24] cannot be interpreted properly by human stakeholders.

Among the root causes of RSs are [20] (1) the structure of the **_prior knowledge_** K; (2) the contents of the **_training set_**; (3) the choice of **_loss function_**; and (4) if the concept extractor is guided by appropriate **_architectural bias_**. Mitigation strategies possibly target one or more of these causes. For instance, *multitask learning* [51] lowers the chance one can achieve high accuracy by confusing concepts, *reconstruction losses* [52] help to disambiguate between visually distinct concepts, and *disentanglement* [53] provides a useful architectural bias. Several other strategies have been proposed [25, 38–40]. Existing solutions, however, are no silver bullet [20]: the only general, sure-proof way of avoiding RSs is supervising concepts (*e.g.*, [54]), which is seldom available and often neglected in learning tasks involving reasoning [25]. By providing easy access to RS-heavy tasks and evaluation protocols, `rsbench` aims to facilitate progress on this challenging open problem.

**Beyond NeSy models and RSs.** While RSs arise naturally in NeSy models, RSs for purely neural architectures are not well-defined as knowledge is not explicitly encoded in such architectures. Nevertheless, several neural models learn concepts either *explicitly* or even *implicitly*, and determining their quality is as important as evaluating RSs in NeSy models, since it provides an indication of potential patterns that the network could end up learning, (*e.g.*, a convolutional filter could learn to detect both a red traffic light and a pedestrian, without disambiguating between the two). Additionally, RSs corrupt the semantics of concept-based explanations extracted in a post-hoc fashion (for NNs) and of model-provided explanations (for CBMs). To this end, we design `rsbench` to evaluate also purely neural models, including gray-box models – such as *concept-bottleneck models* (CBMs) [55] – as well as black-box neural networks (NNs) and neural models involving a pre-processing step given by foundation models, e.g., CLIP [56]. CBMs natively output concept predictions for their decisions, making it possible to directly evaluate their quality using our metrics, at the cost of requiring a modicum of concept-level supervision during training, as shown in Fig. 1 (b). For black-box networks, which only learn concepts implicitly, `rsbench` extracts concept predictions in a post-hoc fashion using TCAV [50], see Fig. 1 (c).

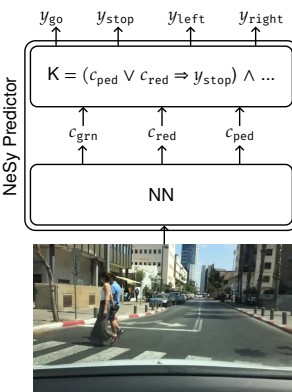

Figure 2: This figure illustrates inference and training in regular NeSy architectures for one `BDD-OIA` example [19]. The input $\mathbf{x}$ is a dashcam image. The model first extracts concepts $\mathbf{c} = (c_{\text{grn}}, c_{\text{red}}, c_{\text{ped}}) \in \{0,1\}^3$ from the image using a neural backbone (NN) and then uses a (differentiable) reasoning layer to infer a vector label $\mathbf{y} = (y_{\text{go}}, y_{\text{stop}}, y_{\text{left}}, y_{\text{right}})$. While the model includes a neural component, the labels depend solely on the extracted concepts. The reasoning layer is aware of prior knowledge K, which encodes constraints like "if a pedestrian or a red light is detected, the prediction must be `stop`."

# 3 The `rsbench` Benchmark Suite

In the following, we outline the L&R tasks and metrics provided by `rsbench`. By construction, RSs do not compromise in-distribution performance, and their worst effects are seen on OOD data. `rsbench` facilitates *constructing novel OOD data sets* by providing a configurable ***data generator*** for each of its tasks (except `BDD-OIA`, cf. Section 3.3). These enable fine-grained control over all details of the training, validation and test splits (like number of examples and percentage allocated to each split, in addition to task-specific settings discussed in the relevant subsection) and the creation of OOD splits, all through a simple YAML configuration file. All tasks are available as Python classes and their knowledge K is supplied in the widely used DIMACS CNF format [57], to support interoperability with model implementations and reasoning packages. In Sections 3.1 to 3.3, for each task, we illustrate a possible reasoning shortcut and its impact on an OOD input.

Table 1 provides an overview of the `rsbench` L&R tasks, breaking them down into relevant properties, namely whether they: include a *data generator* (GEN); allow users to create (✓) or provide ready-made (✅) *out-of-distribution* splits (OOD); allow users to create (✓) or provide ready-made (✅) data suitable for *continual learning* (CONL) [17]; have complex inputs, making it difficult to extract concepts (*e.g.*, of different objects) separately (CPLX $\mathbf{x}$); require complex reasoning when using the default knowledge (CPLX K); by default use knowledge that is intrinsically *ambiguous*, *i.e.*, it yields RSs even if the training set contains all possible combinations of concepts and labels (AMB K). A task involves complex inputs (CPLX $\mathbf{x}$) when it requires processing semi-realistic visual scenes with multiple objects for concept extraction (*e.g*, `Kand-Logic`, `SDD-OIA`). It involves complex reasoning (CPLX K) when inference requires handling interrelated concepts or multi-step reasoning. For instance, `BDD-OIA` and `SDD-OIA` require inferring 4 actions from 20 interrelated concepts (*e.g.*, traffic lights of different colors, presence of pedestrians), where some concepts are mutually exclusive (*e.g.*, traffic lights can't be green and red simultaneously).

Fig. 2 illustrates how a NeSy architecture (DPL) operates on a `rsbench` L&R task (`BDD-OIA`).

## 3.1 Arithmetical Tasks and Data Sets

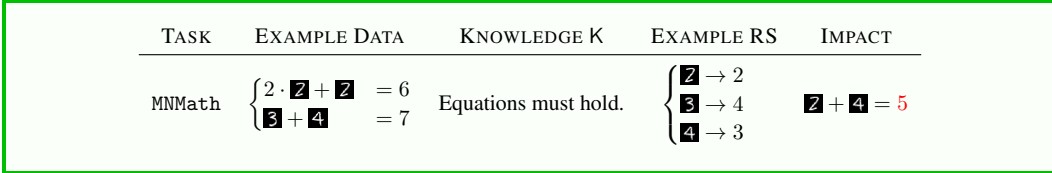

MNAdd [14] is the quintessential benchmark for evaluating reasoning in NeSy AI [7, 12, 38, 58–62]. The goal is to infer the sum of $k \geq 2$ MNIST [63] digits, provided knowledge encoding the rule of summation, that is, $K := (y = \sum_{j=1}^{k} c_j)$. *E.g.*, given $\mathbf{x} =$ `3 4` the model should predict $y = 7$. Despite its simplicity, MNAdd highlights a clear performance gap between pure neural baselines and NeSy architectures [14, 38]. RSs arise when we can infer the correct sum using the wrong digits. This can occur due to commutativity (*e.g.*, `3 /` and `/ 3` both sum to 4) or incomplete training data (*e.g.*, in absence of other training examples, knowing that `2 3` sum to 5 is insufficient to discriminate

the intended sum from $0 + 5$, $1 + 4$, $4 + 1$, and $5 + 0$). If the training set covers all sums in $\{0, \ldots, 18\}$, `MNAdd` only exhibits RSs of the first kind, which can be avoided by processing the two input digits separately [20]. For a detailed break down of individual components (inputs, labels, etc.), see Appendix C.7 and follow-up sections.

**Task:** We introduce `MNMath`, a novel *multi-label* extension of `MNAdd` in which the goal is to predict the result of a system of equations of MNIST digits. *E.g.*, given knowledge $\mathsf{K} = (y_1 = 2c_1 + c_2) \land (y_2 = c_3 + c_4)$ encoding a system of two equations and an input $\mathbf{x} = $ , a model trained to predict $\mathbf{y} = (6, 7)$ can learn to systematically map ▨ to 4 and ▨ to 3, resulting in incorrect down-stream decisions, as in the example above. In `MNMath`, the knowledge consists of a system of equations. The input is a single $28k \times 28$ image, obtained by concatenating $k$ `MNIST` images, each representing a handwritten digit in the equations. The concepts are $k$ categorical variables, one for each digit, and the label encodes the result of the equation system. The key feature of `MNMath` is that, besides requiring more complex reasoning, it comes with a data generator tailored for generating OOD splits and challenging learning scenarios. It allows to change the number of equations and digits in each equation, and to define additional operations.

**Task:** To facilitate comparison with existing mitigation algorithms, `rsbench` supplies also `MNAdd-Half` and `MNAdd-EvenOdd`, two variants of `MNAdd` with guaranteed RSs that have been used in the literature [25, 17]. Both restrict the digits available for training to a subset of combinations – `MNAdd-Half` focuses on certain combinations of digits in $\{0, \ldots, 4\}$, while `MNAdd-EvenOdd` contains either only even or only odd digits – guaranteeing the model cannot avoid RSs even when processing inputs separately. In contrast with `MNAdd-Half`, however, it naturally lends itself to OOD evaluation, with the even digits in one domain and the odd digits the second one, and to multitask and continual settings. For these datasets, the knowledge is based on the sum operation. The input is a single $56 \times 28$ image, created by concatenating 2 `MNIST` images, each representing an operand in the sum. The concepts consist of 2 categorical variables, one for each digit, and the label represents the sum of these digits.

## 3.2 Logical Tasks and Data Sets

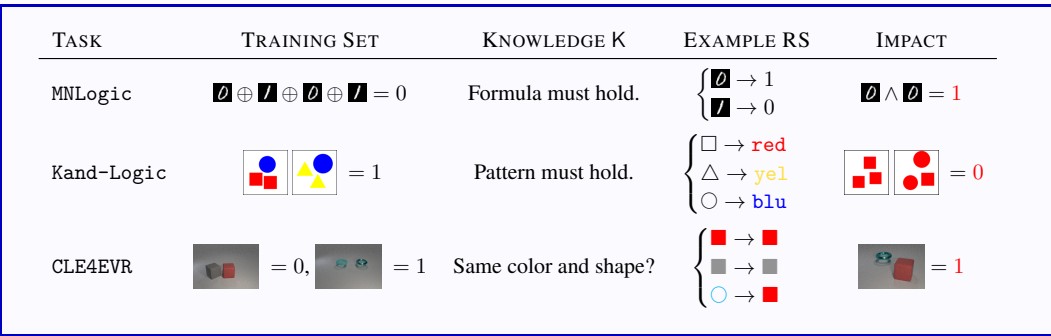

**Task: `MNLogic`.** RSs arise whenever the knowledge $\mathsf{K}$ allows deducing the right label from multiple configurations of concepts. This form of non-injectivity is a standard feature of most logic formulas, and in fact formulas as simple as the XOR are riddled by RSs [20]. *I.e.*, if $\mathsf{K} = (c_1 \oplus c_2 \Leftrightarrow y)$, where $\oplus$ denotes the XOR operator, a model that maps ▨ to 1 and ▨ to 0 classifies all inputs perfectly. To probe the pervasiveness of RSs, we introduce `MNLogic`, a logical analogue to `MNAdd` [14] in which inference is driven by a *random logic formula* $\varphi$. Specifically, an input $\mathbf{x}$ is the concatenation of $k \geq 2$ images of zeros and ones sampled from MNIST [63] representing the truth value of $k$ bits $\mathbf{c}$, and its ground-truth label $y$ encodes whether it satisfies $\varphi$ or not, *i.e.*, $\mathsf{K} = (\varphi \Leftrightarrow y)$. The formula $\varphi$ is a random $\ell$-CNF, *i.e.*, a conjunction of $m$ clauses (disjunctions), each of which contains $\ell$ out of the $k$ bits and their negations. For instance, $x = $ ▨▨ encodes the bits $\mathbf{c} = (0, 1)$ and if $\varphi = (c_1 \lor c_2) \land (c_1 \lor \neg c_2)$ it is labeled as $y = 0$. In `MNLogic`, the knowledge consists of a logical formula $\varphi$. The input is a single $28k \times 28$ image, created by concatenating $k$ `MNIST` images, each representing either a true atom (▨) or a false atom (▨) in the formula. The concepts are $k$ categorical variables, one for each atom, and the label encodes the outcome of the logical formula. The `MNLogic` generator allows users to specify the number of digits $k$, the number of clauses $m$ and their length $\ell$, and to supply a custom formula, and takes care of constructing all training, validation, and test splits. It also avoids trivial data by ensuring each clauses is neither a tautology nor a contradiction.

**Task:** `Kand-Logic` [25] is a task – inspired by Wassily Kandinsky's paintings and [64] – that requires simple (but non-trivial) perceptual processing and relatively complex reasoning. In the simplest case, each input $x = (x_1, x_2)$ consists of two $64 \times 64$ images, each depicting three geometric primitives with different shapes ($\square$, $\triangle$, $\bigcirc$) and colors (red, blue, yellow). The goal is to predict whether $x_1$ and $x_2$ fit the same predefined *logical pattern* or not. Let each $x_i$ contain three primitives $x_{i,1}$, $x_{i,2}$, $x_{i,3}$ with two concepts each: shape $\mathtt{sha}(x_{ij})$ and color $\mathtt{col}(x_{ij})$. The pattern is built out of predicates like "all primitives in the image have a different color", "all primitives have the same color", and "exactly two primitives have the same shape", formally:

$$\mathtt{diffcol}(x_i) = \bigwedge_{j \neq j'}(\mathtt{col}(x_{ij}) \neq \mathtt{col}(x_{ij'})), \quad \mathtt{samecol}(x_i) = \bigwedge_{j \neq j'}(\mathtt{col}(x_{ij}) = \mathtt{col}(x_{ij'})),$$

and $\mathtt{twosha}(x_i) = \neg\mathtt{samesha}(x_i) \wedge \neg\mathtt{diffsha}(x_i)$. For instance, the default pattern [25] $\mathtt{patt}(x_i)$ is "all images include either the same number of primitives with the same color, or the same number of primitives with the same shape", or equivalently:

$$(\mathtt{diffcol}(x_1) \wedge \mathtt{diffcol}(x_2)) \vee (\mathtt{twocol}(x_1) \wedge \mathtt{twocol}(x_2)) \vee (\mathtt{samecol}(x_1) \wedge \mathtt{samecol}(x_2))$$
$$\vee (\mathtt{diffsha}(x_1) \wedge \mathtt{diffsha}(x_2)) \vee (\mathtt{twosha}(x_1) \wedge \mathtt{twosha}(x_2)) \vee (\mathtt{samesha}(x_1) \wedge \mathtt{samesha}(x_2))$$

An input $x$ is positive if and only if it satisfies the pattern, *i.e.*, $\mathsf{K} = (\mathsf{Y} \Leftrightarrow \mathtt{patt}(x_1, x_2))$. Unlike `MNLogic`, in `Kand-Logic` each primitive has multiple attributes that cannot easily be processed separately. This means that RSs can easily, *e.g.*, confuse shape with color when either is sufficient to entail the right prediction, as in the example above. `rsbench` provides the data set used in [25] (3 images per input with 3 primitives each) and a generator that allows configuring the number of images and primitives per input and the pattern itself.

**Task:** `CLE4EVR` [17] focuses on logical reasoning over three-dimensional scenes, inspired by `CLEVR` [65] and `CLEVR-HANS` [66]. Among these, `CLEVR` is tailored for visual-question answering and `CLEVR-HANS` to contain confounding factors at the input level, to make *shortcuts* arise [67]. Both of them typically provide models with exhaustive concept-level supervision, obscuring whether RSs are present without it. Our `CLE4EVR` constitutes a simplified version where RSs can be easily determined. Each input image $x$, of size $240 \times 320$, contains a variable number of objects differing in size (3 possible values), shape (10), color (10), material (2), position (real), and rotation (real), and the goal is to determine whether the objects satisfy a pre-specified condition $\varphi$ that depends on all discrete attributes of the objects in the scene. The default knowledge $\mathsf{K}$ is designed to induce RSs [17]: it asserts that an image $x$ is positive iff at least two objects $x_i$ and $x_j$ have the same color and shape, *i.e.*, $\exists i \neq j . (\mathtt{sha}(x_i) = \mathtt{sha}(x_j)) \wedge (\mathtt{col}(x_i) = \mathtt{col}(x_j))$. When all possible colors and shapes are observed, the only RSs `CLE4EVR` is affected by are those in which the attributes of first object are attributed to the second one. However, if the training set includes only *some* combinations – *e.g.*, pink rings and gray spheres are never observed together – the model can collapse different shapes and colors [17]. Hence, even when objects are processed separately (*e.g.*, via Faster-RCNN embeddings), the model can confuse colors and shapes with one another, *e.g.*, it can mistake blue cones for red pyramids and vice versa. Occlusion further complicates the picture, complicating both perception and reasoning. As above, the generator allows to customize the number of objects per image, the knowledge, and whether occlusion is allowed.

### 3.3 High-stakes Tasks and Data Sets

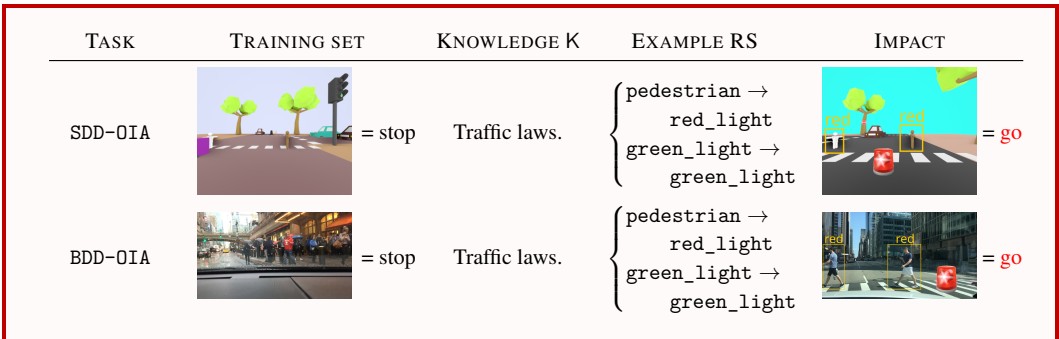

**Task:** `BDD-OIA` [19] is a multi-label autonomous driving task for studying RSs in real-world, *high-stakes* scenarios. The goal is to infer what actions out of {forward, stop, left, right} are safe

depending on what objects (*e.g.*, cars, traffic signs) are present in an input dashcam image. The knowledge K establishes that, *e.g.*, it is not safe to move `forward` if there are pedestrians on the road, based on a set of 21 binary concepts indicating the presence of different obstacles on the road. The constraints specify conditions for being able to proceed (`green_light` $\vee$ `follow` $\vee$ `clear` $\Rightarrow$ `forward`), stop (`red_light` $\vee$ `stop_sign` $\vee$ `obstacle` $\Rightarrow$ `stop`), and for turning left and right, as well as relationships between actions (*e.g.*, `stop` $\Rightarrow$ `¬forward`). Input images, of size $720 \times 1280$, come with concept-level annotations, making it possible to assess the quality of the learned concepts. The dataset comprises $16,082$ training examples, $2,270$ validation examples and $4,572$ test examples. Common RSs allow to, *e.g.*, confuse `pedestrians` with `red_lights`, as they both imply the correct (`stop`) action for all training examples [20].

**Task:** SDD-OIA. BDD-OIA is not suitable for systematically evaluating RSs out-of-distribution, where they show the highest impact. With `rsbench`, we fill this gap by introducing SDD-OIA, a synthetic replacement for BDD-OIA that comes with a fully configurable *data generator*, enabling fine-grained control over what labels, concepts, and images are observed and the creation of OOD splits. In short, SDD-OIA shares the same classes, concepts and (by default) knowledge as BDD-OIA, but the images are 3D traffic scenes modelled and rendered using Blender [68] as $469 \times 387$ RGB images. Images are generated by first sampling a desired label $\mathbf{y}$, then picking concepts $\mathbf{c}$ that yield that label, and then rendering an image $\mathbf{x}$ displaying those concepts. This allows to easily control what concepts and labels should appear in all data splits, which in turn determine what kinds of RSs can be learned. The complete data generation process is described in Appendix C.11.

In Section 4, we showcase SDD-OIA by implementing an OOD autonomous ambulance scenario [20] in which the vehicle is allowed to cross red lights in case of an `emergency`. Formally, this requires altering the prior knowledge by introducing a new `emergency` variable that conditions the traffic rules, that is, ($\neg$`emergency` $\implies$ original rule for `stop`) $\wedge$ ($\neg$`emergency` $\implies$ alternative rule for `stop`), and similarly for `turn_left` and `turn_right`. We specifically test this scenario in Section 4. Naturally, other challenging OOD scenarios can be created.

### 3.4 Metrics for Reasoning Shortcuts

**Model-level metrics.** `rsbench` facilitates assessing learned models by implementing several metrics for label and concept predictions – including accuracy and $F_1$ score – as well as metrics for RSs. First, `rsbench` provides concept-level confusion matrices, which show how well the predicted concepts $\mathbf{c} = (c_1, \ldots, c_k)$ recover the annotations $\mathbf{c}^* = (c_1^*, \ldots, c_k^*)$ and are essential for visualizing and spotting RSs, as can be seen in Table 3. Second, it implements ***concept collapse*** $\mathsf{Cls}(C)$, which measures to what extent the learned concepts mix distinct ground-truth concepts. Given a concept confusion matrix $C \in [0,1]^{m \times m}$, where $m$ is the size of the confusion matrix (*e.g.*, $m = 2^k$ when all ground-truth concepts are observed), it is defined as $\mathsf{Cls}(C) = 1 - p/m$, where $p = \sum_{j=1}^{m} \mathbb{1}\{\exists i \, . \, C_{ij} > 0\}$. High collapse shows that the model tends to use fewer concepts to solve the task, making it useful for diagnostics. Vice versa, a lower concept collapse may indicate that the RS is densely activating all concepts. Concept collapse is not trivial to implement because not all $2^k$ ground-truth concept combinations may appear in the test set (especially when $k$ is large, see Appendix A for details), so `rsbench` provides ready-made implementations for all its tasks.

**Task-level metrics.** RSs arise due to a complex interaction between prior knowledge and training data (cf. Section 2) making it difficult to assess *a priori* which L&R tasks they affect. Fortunately, it is possible to count how many optimal (deterministic) RSs affect a L&R task [20, 25], as long as this satisfies two technical assumptions.[3] They also provide a closed-form expression for the count that works only when the training set is *exhaustive* (that is, comprises all possible combinations of concepts, like MNAdd) and the concepts are extracted jointly. This makes it possible to ***formally verify*** whether a L&R task can be solved via RSs by checking that the count is larger than 1. This is crucial for anticipating the occurrence of RSs in novel tasks and for iteratively improving task specifications in the design stage. In practice, however, the training set is seldom exhaustive and concepts are often processed separately. While the former issue can be overcome – and in fact, we provide a closed-form solution in Appendix A.3 – the latter is more challenging.

---

[3]Specifically, ***invertibility*** (**A1**) and ***determinism*** (**A2**), meaning that it is always possible to recover the unique ground-truth concept vector underlying any input and that the knowledge K maps any concept configuration to a unique label, respectively; see [20] for an in-depth discussion.

Table 2: Results on `MNAdd-EvenOdd`

|  | $F_1(Y)(\uparrow)$ | $F_1(C)(\uparrow)$ | $Cls(C)(\downarrow)$ |
|---|---|---|---|
| DPL | $0.94 \pm 0.04$ | $0.06 \pm 0.08$ | $0.61 \pm 0.08$ |
| LTN | $0.66 \pm 0.10$ | $0.05 \pm 0.06$ | $0.70 \pm 0.01$ |
| CBM$^\dagger$ | $0.89 \pm 0.13$ | $0.44 \pm 0.07$ | $0.09 \pm 0.09$ |
| NN$^*$ | $0.57 \pm 0.38$ | $0.07 \pm 0.03$ | $0.29 \pm 0.34$ |
| CLIP$^*$ | $0.62 \pm 0.14$ | $0.04 \pm 0.01$ | $0.48 \pm 0.17$ |

Table 3: (Left) DPL and (Right) NN concept confusion matrices for `MNAdd-EvenOdd`.

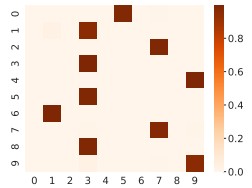 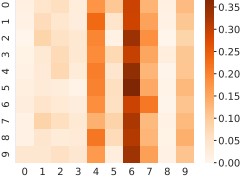

Table 4: Results on `Kand-Logic`

|  | $F_1(Y)(\uparrow)$ | $F_1(C)(\uparrow)$ | $Cls(C)(\downarrow)$ |
|---|---|---|---|
| DPL | $0.87 \pm 0.15$ | $0.25 \pm 0.09$ | $0.69 \pm 0.04$ |
| LTN | $0.77 \pm 0.09$ | $0.35 \pm 0.04$ | $0.00 \pm 0.01$ |
| CBM$^\dagger$ | $0.36 \pm 0.04$ | $0.59 \pm 0.01$ | $0.00 \pm 0.01$ |
| NN$^*$ | $0.72 \pm 0.08$ | $0.33 \pm 0.01$ | $0.17 \pm 0.01$ |
| CLIP$^*$ | $0.99 \pm 0.01$ | $0.32 \pm 0.01$ | $0.00 \pm 0.01$ |

Table 5: Performance on `SDD-OIA`

|  | $mF_1(Y)(\uparrow)$ | $mF_1(C)(\uparrow)$ | $mCls(C)(\downarrow)$ | $OOD\text{-}mF_1(Y)(\uparrow)$ |
|---|---|---|---|---|
| DPL | $0.80 \pm 0.01$ | $0.49 \pm 0.03$ | $0.86 \pm 0.04$ | $0.62 \pm 0.09$ |
| LTN | $0.82 \pm 0.04$ | $0.46 \pm 0.04$ | $0.81 \pm 0.02$ | $0.72 \pm 0.06$ |
| CBM$^\dagger$ | $0.60 \pm 0.12$ | $0.61 \pm 0.04$ | $0.68 \pm 0.06$ | $0.45 \pm 0.05$ |
| NN$^*$ | $0.93 \pm 0.18$ | $0.44 \pm 0.02$ | $0.43 \pm 0.28$ | $0.47 \pm 0.19$ |
| CLIP$^*$ | $0.90 \pm 0.09$ | $0.43 \pm 0.04$ | $0.23 \pm 0.02$ | $0.81 \pm 0.06$ |

`rsbench` addresses it by implementing a practical counting algorithm, named `countrss`, that leverages *automated reasoning* [49] techniques. In a nutshell, each optimal RS can be viewed as a linear mapping $\mathbf{c} = A\mathbf{c}^*$ that maps ground-truth to predicted concepts under the *constraint* that these yield the correct label $\mathbf{y}^*$ for all training examples. The problem thus boils down to counting how many matrices $A \in \{0,1\}^{k \times k}$, where $k$ is the number of concepts,[4] satisfy this constraint. Given the exponential (in $k$) number of candidates, `countrss` relies on state-of-the-art *model counting* solvers [49, 69] for efficiency. That is, we encode the above *constraint* as a propositional logic formula (see Appendix A for the exact encoding), such that each model (solution) represents a distinct RS. `countrss`, based on `PyEda` [70], works for all L&R tasks that satisfy the necessary technical assumptions, including all those in `rsbench` except `BDD-OIA`, and supports both exact [71] and, for the more complex tasks, approximate counting [72].

We showcase `countrss` by evaluating the impact of the amount of training examples on the RS count for two instances of `MNLogic`: AND (with $K = (y \Leftrightarrow c_1 \wedge c_2 \wedge c_2)$) and XOR ($K = (y \Leftrightarrow c_1 \oplus c_2 \oplus c_3)$). When the training set is exhaustive, AND admits 6 RSs and XOR 24, proving that symmetries in the XOR function make it latter more ambiguous in this case, as stated in Section 3.2. The number of RSs grows drastically when we only provide a single example, as it becomes easier to predict all labels correctly while confusing concepts, with the XOR presenting 192 RSs. For the AND, the count depends on whether the single ground truth label is positive or negative, the number of RSs growing to 48 and 336, respectively. This highlights how even simple formulas can be affected by RSs and that these depend crucially on the available data, as expected, and that `countrss` can anticipate the occurrence of RSs in L&R tasks without the need for training any model.

## 4 Evaluating RSs and Concept Quality with `rsbench`

`rsbench` is meant to be a general framework for evaluating the impact of RSs and concept quality in *any* machine learning model. We showcase this by evaluating *five* different architectures on three L&R tasks, one per "flavor", namely `MNAdd-EvenOdd`, `Kand-Logic`, and `SDD-OIA`.

We consider two state-of-the-art NeSy models: DeepProbLog (`DPL`) [14] and Logic Tensor Networks (`LTN`) [34]. Both comprise a neural network module to extract concepts $\mathbf{c}$ for every input $\mathbf{x}$, which are later used to predict labels $\mathbf{y}$ according to the knowledge $K$ (Fig. 1(a)). These predictions are done according to a probabilistic logic semantic in `DPL` and by using fuzzy logic in `LTN`. As stated in Section 2, we also experiment with purely neural models, evaluating the quality of the concepts they learn on `rsbench`. Specifically, we employ CBMs (Fig. 1(b)), black-box NNs and CLIP (Fig. 1(c)). In our analysis, we investigate directly the bottleneck layer for CBM, where concepts are expected to be learned, and adopt TCAV for NN and CLIP.

---

[4]More precisely, $k$ is the number of bits required for the one-hot encoding of the concept vector $\mathbf{c}$.

We evaluate macro $F_1$ for predicted labels and concepts, denoted as $F_1(Y)$ and $F_1(C)$, respectively, and concept collapse $\text{Cls}(C)$ (see Appendix A for the exact definition). We report the mean and standard deviation over 10 random seeds. We train all models by maximum likelihood on the labels, as customary in NeSy and for neural baselines. Notice that, CBM without annotated concepts would be equivalent to NN, therefore we supervise a handful of concepts, as customary [55, 22, 23, 73]. Specifically, we supervise a total of 100 examples for MNAdd-EvenOdd (only for the digits 3, 4, 8 and 9); 20 examples for Kand-Logic (for red, □, and ○); and $\sim 700$ examples for SDD-OIA on the high-stakes concepts red_light, green_light, car, person, rider, other_obstacle, stop_sign, right_green_light, and left_green_light. All details about the losses, architectures, metrics, and model selection procedure we use are reported in Appendix B.

**All tasks succeed in inducing RSs across all models.** The results in Tables 2, 4 and 5 show that all models attain medium-to-high $F_1(Y)$ on the three benchmarks, meaning the labels *can* be predicted accurately, with the following exceptions: On MNAdd-EvenOdd LTN, NN, and CLIP show medium-to-high variance (10%, 38%, and 14%, respectively); On Kand-Logic, LTN, CBM, and NN reach suboptimal performance on (around 77%, 36%, and 72%, respectively); On SDD-OIA, CBM scores only 60% $F_1(Y)$. We attribute the subpar $F_1(Y)$ score of CBM to the fact that their top linear layer is not expressive enough to accurately infer the label from the concept bottleneck in more complex tasks like Kand-Logic and SDD-OIA. Despite these variations in prediction performance, *all models show overall low concept quality*, as measured by $F_1(C)$. Even CBM, despite receiving concept supervision, fare below 60%.

**Understanding concept quality with rsbench.** The high rate of $\text{Cls}(C)$ in all tasks (except for LTN in Kand-Logic) suggests that NeSy models tend mix concepts together [40]. The left-most confusion matrix in Table 3 shows that in MNAdd-EvenOdd, DPL uses roughly half of the available digits to solve the task with high $F_1(Y)$. In contrast, CBM experiences less collapse due to concept supervision. NN and CLIP also yield overall low collapse (an exception being CLIP in MNAdd-EvenOdd), and in fact the right-most confusion matrix in Table 3 shows that NN in MNAdd-EvenOdd activates densely most concepts. We point out that, however, lower collapse for NN and CLIP stems also from TCAV, which can introduce noise in the extracted concepts. Due to space constraints, we report a detailed analysis of this phenomenon in the supplementary material.

**Generators enable measuring the OOD impact of RSs.** For SDD-OIA we leverage the generator to evaluate an OOD setting where the same concepts are used with a different knowledge $\text{K}_{\text{OOD}}$ (reported in Section 3.3). *We observe that all models suffer a visible drop in OOD* $\text{mF}_1(Y)$ *performance*, as expected: DPL drops by 18%, LTN by 10%, and CBM by $\sim 15\%$. NN is the most affected, with average 46% difference, while CLIP is the most resilient with only 9% drop.

## 5   Discussion and Conclusion

We introduced rsbench, an integrated benchmark suite for systematic evaluation of RSs and concept quality in tasks requiring learning and reasoning. While existing benchmark suites [44] neglect RSs altogether, rsbench supplies datasets for various RS-heavy tasks and corresponding ready-made data generators for evaluating OOD and continual learning scenarios. At the same time, rsbench provides formal verification and evaluation routines for assessing how much RSs and concept quality affect each task. Our experiments showcase how rsbench enables practitioners to easily investigate the impact of RSs on several existing or future deep learning architectures.

RSs are also connected to the more general problem of learning high-level concepts from data, *aka* symbol grounding [74, 75]. Interpretable concepts play an increasingly central role as a *lingua franca* in explainable AI (XAI) [21, 76] for both *post-hoc* [50, 77–79] and *ante-hoc* [55, 73, 22, 80, 81, 24] explanations of model decisions. As with RSs, a central question is whether the concepts encode the intended semantics [82, 83]. rsbench can be used to benchmark precisely whether learned concepts satisfy this condition. Furthermore, it can also benefit new research in mechanistic interpretability [84–88], specifically for studying challenging scenarios in which deriving a high-level explanation of neural networks behavior is complicated by poor concept semantics.

Another related topic is *identifiability* of latent concepts, which is studied in independent component analysis [89] and causal representation learning [53, 90, 91]. rsbench can be readily used to empirically assess identification of latent concepts with only label supervision [92–94].

**Broader Impact.** Concepts confused by RSs can lead to poor down-stream decision making. With `rsbench`, we hope to enable research on mitigation strategies for RSs to avoid such consequences. At the same time, `rsbench` might be used to design adversarial attack that exploit or promote RSs and therefore undermine the trustworthyness of ML systems.

**Limitations and Future Work.** While `rsbench` already provides a variety tasks offering different learning and reasoning challenges, we plan to extend it to include variants of other popular reasoning datasets such as Visual Sudoku [95], Raven matrices [96], KANDY [97], CLEVR-HANS [66], and ROAD-R [98] which in their current status do not allow a systematic study of RSs. Implementations of NeSy architectures make use of distinct formalisms and file formats, making it especially challenging to ensure data interoperability. `rsbench` partially addresses this by supplying both Python APIs and CNF specifications – the standard file format for logic formulas in formal verification – for all L&R tasks. In the future, we will build on initiatives like ULLER [99], which promise to provide a unified interface for NeSy architectures. We also plan to improve the scalability of the formal verification algorithm to larger L&R tasks and to leverage as a guide for active learning-based mitigation strategies.

### Acknowledgements

The authors are grateful to Alice Plebe for her valuable guidance on Blender, as well as to the authors of the assets used in `SDD-OIA`, see Appendix C.11.1 for the full list. Funded by the European Union. The views and opinions expressed are however those of the author(s) only and do not necessarily reflect those of the European Union, the European Health and Digital Executive Agency (HaDEA) or the European Research Executive Agency. Neither the European Union nor the granting authority can be held responsible for them. Grant Agreement no. 101120763 - TANGO. PM is supported by the MSCA project GA nř101110960 Probabilistic Formal Verification for Provably Trustworthy AI - PFV-4-PTAI. AV is supported by the "UNREAL: Unified Reasoning Layer for Trustworthy ML" project (EP/Y023838/1) selected by the ERC and funded by UKRI EPSRC. Emile van Krieken was funded by ELIAI (The Edinburgh Laboratory for Integrated Artificial Intelligence), EPSRC (grant no. EP/W002876/1).

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

# A Metrics: Additional Details

## A.1 Model-level Metrics

**Label and Concept Evaluation.** For all datasets, we evaluate the predictions on the labels by measuring the $F_1$-score with macro average. We followed these specifics for `MNAdd-EvenOdd` and `Kand-Logic`.

In `BDD-OIA` and `SDD-OIA`, there are 4 labels and 21 concepts, and to measure the $F_1$ score we adopt a softened metric [19, 100], namely, the mean-$F_1$ score and a mean accuracy. Specifically, we first compute the binary $F_1$-score and accuracy for each concept $C_i$ and then average them:

$$\mathrm{mF}_1(Y) = \frac{F_1(\text{forward}) + F_1(\text{stop}) + F_1(\text{left}) + F_1(\text{right})}{4} \tag{1}$$

Similarly, for the concepts, we perform the following:

$$\mathrm{mF}_1(C) = \frac{F_1(\text{green\_light}) + \cdots + F_1(\text{right\_follow})}{21} \tag{2}$$

**Concept Collapse.** For all datasets, to measure the *Concept collapse* $\mathsf{Cls}(C)$, we first compute the confusion matrix. Here, we provide additional details when not all ground-truth concepts $\mathbf{C}^*$ appear in the test set. To this end, it is desirable to mention how the confusion matrix is extracted.

Let $\mathcal{C}^* \subseteq \{0,1\}^k$ be the subset of concepts vectors appearing in the test set, and $\mathcal{C}$ be the subset of concepts vectors predicted by the model for inputs in the test set, *e.g.*, taking the MAP estimate. Notice that both $|\mathcal{C}|$ and $|\mathcal{C}^*|$ are below or equal to $2^k$. To evaluate the confusion matrix in this multilabel setting, the set of labels is determined by first converting each binary string to its integer value, *e.g.*, $(0,1,1) \mapsto 3$. Let $\mathcal{F}(\mathcal{C})$ and $\mathcal{F}(\mathcal{C}^*)$ be the two subsets converted to categorical values from $\mathcal{C}$ and $\mathcal{C}^*$, respectively. From $\mathcal{F}(\mathcal{C})$ and $\mathcal{F}(\mathcal{C}^*)$ we obtain the confusion matrix $C$ using the `scikit-learn` library [101]. In this case, the output would be a matrix $C \in [0,1]^{m \times m}$, where $m = |\mathcal{F}(\mathcal{C}) \cup \mathcal{F}(\mathcal{C}^*)|$, where categorical values not appearing in $\mathcal{F}(\mathcal{C}^*)$ will give empty rows, *i.e.*, $C_{i,:} = (0, \ldots, 0)^\top$, for all $i \notin \mathcal{F}(\mathcal{C}^*)$. Following the previous definition, we obtain that:

$$p = \sum_{j=1}^{m} \mathbb{1}\{\exists i \,.\, C_{ij} > 0\} = |\mathcal{F}(\mathcal{C})| \tag{3}$$

Then, collapse can be evaluated as before:

$$\mathsf{Cls}(C) = 1 - \frac{p}{m} \tag{4}$$

Notice that when $m = 2^k$ the form reduces to the one discussed in the main text.

For `Kand-Logic`, we took the concept for each geometric figure, using a 6-dimensional one-hot encoding for shape (3) and color (3), and computed the collapse after converting this base 3 representation to a base 10 integer. `rsbench` provides a way to compute the collapse for shape and color separately. In this case, we compute the confusion matrix as is without the need for a conversion. We threat the digits predictions in `MNAdd-EvenOdd`, in the same way.

To compute the $\mathsf{Cls}(C)$ for `BDD-OIA` and `SDD-OIA`, we convert the binary 21-concept prediction to an integer and compute the collapse. `rsbench` also allows computing the collapse for each concept associated with corresponding categories (e.g., `move_forward`, `stop`, `turn_left`, `turn_right`) in the same manner.

## A.2 The Impact of TCAV

We extract concepts from neural networks by leveraging the TCAV post-hoc explainer [50]. In essence, TCAV acts as a linear probe: for each concept, it trains a binary linear classifier using the network's embeddings (typically from the second-to-last layer) as inputs and the concept's annotations as targets, distinguishing between when a concept is present ($x_c$) and when it is absent ($x_{\neg c}$). From

this classifier, we extract the Concept Activation Vector (CAV), which corresponds to the weights of the linear decision boundary, denoted as $\mathbf{v}_{\text{CAV}}$.

To assess whether a concept is present, we use the TCAV score, which checks whether the model's prediction aligns positively with the CAV by computing:

$$\frac{\partial f(x_i)}{\partial h(x_i)} \cdot \mathbf{v}_{\text{CAV}}, \tag{5}$$

where $\frac{\partial f(x_i)}{\partial h(x_i)}$ represents the gradient of the model's output with respect to the embedding space $h(x_i)$, and $\cdot$ denotes the dot product. A positive score suggests that the concept is contributing to the model's prediction.

We employed the TCAV score to determine the presence of concepts in both `NN` and `CLIP`, as discussed in Section 4.

One issue is that TCAV is not always reliable, depending on the task at hand, meaning that it might mis-predict the concepts learned by the model, simply because these concepts may not be linearly separable in the embedding space. This occurs even when the linear classifiers perform well on held-out data.

This can lead to overestimating concept collapse, as noted in Section 4, and to underestimating the quality of implicitly learned concepts. Possible remedies, which we plan to implement as we develop `rsbench` further, include replacing TCAV with more advanced techniques from mechanistic interpretability [78, 85].

### A.3 Task-level Metrics

To properly understand how the `countrss` works and how it is possible to count the number of RSs we need to introduce technical details on what assumptions and conditions are required. We report in this section an overview of the theoretical material in [20, 25], precisely meant to explain comprehensively the counting of RSs. Additional details about derivations and proofs for RSs characterization can be found in the references above.

To this end, we need to (*i*) introduce the functional form of NeSy predictors along with their training objective and the optimality condition, and (*ii*) the data generation process and the notion of intended semantics. (*iii*) By leveraging two simplifying assumptions, it is possible to derive a formula for counting the number of optimal solutions (including RSs).

**Neuro-Symbolic models and Learning Objective.**  Without loss of generality, we analyze Neuro-Symbolic models leveraging Probabilistic Logic approaches [16, 14, 9] of the form:

$$p_\theta(\mathbf{y} \mid \mathbf{x}; \mathsf{K}) = \sum_{\mathbf{c} \in \{0,1\}^k} p(\mathbf{y} \mid \mathbf{c}; \mathsf{K}) p_\theta(\mathbf{c} \mid \mathbf{x}) \tag{6}$$

The ***perception*** step is performed via $p_\theta(\mathbf{c} \mid \mathbf{x})$, computing a high-level conceptual description of a low-level input, usually implemented as a neural network; and the ***reasoning*** step is performed by $p(\mathbf{y} \mid \mathbf{c}; \mathsf{K})$, which infers a prediction $\mathbf{y}$ based on the high-level description $\mathbf{c}$ and prior knowledge $\mathsf{K}$.

The learning objective for models of this form is given by the maximization of the likelihood on data a set $\mathcal{D} = \{(\mathbf{x}, \mathbf{y})\}$. We denote with (**D1**) the condition to which NeSy models attain the optimum of likelihood that is $\theta^* \in \arg\max_\theta \sum_{(\mathbf{x},\mathbf{y}) \in \mathcal{D}} \log p_\theta(\mathbf{y} \mid \mathbf{x}; \mathsf{K})$.

**Data Generation Process and Intended Semantics.**  Similar to [20], we assume data are distributed according to a ground-truth distribution $p^*(\mathbf{x}, \mathbf{y}; \mathsf{K})$. There exist $k$ latent, *ground-truth concepts* $\mathbf{c}^* \in \{0,1\}^k$ drawn from an unobserved distribution $p(\mathbf{c}^*)$, where input variables $\mathbf{x} \in \mathbb{R}^n$ are distributed according to the conditional $p(\mathbf{x} \mid \mathbf{c}^*)$. Latent concepts also generate the label $\mathbf{y}$ according to distributions $p(\mathbf{x} \mid \mathbf{c}^*; \mathsf{K})$. The overall distribution on the observed inputs and output labels is given by $p(\mathbf{x}, \mathbf{y}; \mathsf{K}) = \mathbb{E}_{\mathbf{c}^* \sim p^*(\mathbf{c}^*)} p(\mathbf{x} \mid \mathbf{c}^*) p(\mathbf{y} \mid \mathbf{c}; \mathsf{K})$. The conditional distribution $p(\mathbf{c}^* \mid \mathbf{x})$ describes how concepts are distributed according to the input.

Based on this, we denote with (**D2**) the condition for which the learned concepts possess the intended semantics, that is $p_\theta(\mathbf{c} \mid \mathbf{x}) \equiv p(\mathbf{c}^* \mid \mathbf{x})$.

**Reasoning Shortcuts are Optimal Solutions with Unintended Semantics.** Having specified the meaning of intended concepts (**D2**) and the optimality condition for NeSy models (**D1**), does **D1** $\implies$ **D2**, even in the limit of infinite data? Attaining optimality with NeSy models is insufficient to learn concepts with the intended semantics [20], making the implication not to hold in general.

It is possible to detect the presence of RSs before-hand, leveraging two technical assumptions:

**A1 Invertibility**, the ground-truth map from input to concept is given by a function $f^* : \mathbf{x} \mapsto \mathbf{c}^*$, *i.e.*, $p(\mathbf{c}^* \mid \mathbf{x}) = \mathbb{1}\{\mathbf{c}^* = f^*(\mathbf{x})\}$;

**A2 Determinism**, the labels are uniquely determined by the knowledge K and concepts by a function $\beta_{\mathsf{K}} : \mathbf{c} \mapsto \mathbf{y}$ underlying the knowledge, *i.e.*, $p(\mathbf{y} \mid \mathbf{c}, \mathsf{K}) = \mathbb{1}\{\mathbf{y} = \beta_{\mathsf{K}}(\mathbf{c})\}$.

Intuitively, **A1** indicates that ground-truth concepts are determined uniquely from the input, *i.e.*, ground-truth concepts can be read from the input variables. This means that **D2** reduces to obtain the function $f^*$ via the model $p_\theta(\mathbf{c} \mid \cdot)$. On the other hand, **A2** subtends the fact that for knowledge K there is only one $\mathbf{y}$ that complies with concepts $\mathbf{c}$, that is $(\mathbf{c}, \mathbf{y}) \models \mathsf{K}$ and $(\mathbf{c}, \mathbf{y}') \models \mathsf{K}$, *if and only if* $\mathbf{y}' = \mathbf{y}$. This also means that given the concepts and the knowledge, only one label vector is associated with them.

We indicate with $\mathrm{supp}(\mathbf{c}^*)$ the support of the ground-truth concepts distribution $p(\mathbf{c}^*)$ and with $\mathcal{A}$ the space of all maps $\alpha : \{0,1\}^k \to \{0,1\}^k$, mapping one concept vector to another. Based on this, we can derive a count for optimal NeSy models of a particular form:

**Theorem 1** (Misspecification of NeSy models [20]). *Under **A1** and **A2**, the number of models of the form $p_\theta(\mathbf{c} \mid \mathbf{x}) = \mathbb{1}\{\mathbf{c} = f_\theta(\mathbf{x})\}$, with $f_\theta = (\alpha \circ f^*)$, attaining maximum likelihood amounts to:*

$$\sum\nolimits_{\alpha \in \mathcal{A}} \mathbb{1}\Big\{ \bigwedge\nolimits_{\mathbf{c} \in \mathrm{supp}(\mathbf{c}^*)} (\beta_{\mathsf{K}} \circ \alpha)(\mathbf{c}) = \beta_{\mathsf{K}}(\mathbf{c}) \Big\} \tag{7}$$

In a nutshell, the theorem proves that the number of alternative solutions to the ground-truth one $f^*$ is given by those maps $\alpha \in \mathcal{A}$ that *map ground-truth concepts to valid alternatives for label predictions*. When the number is above 1, RSs are present in the learning problem. The formula offers a principled way to count RSs in practice and allows to design tasks where RSs are present and we show a practical implementation with `countrss`.

**Explicit count of optimal solutions.** Marconato *et al.* [20] showed that it is possible to derive an analytical expression for the total number of optimal solutions appearing in Eq. (7). This requires making assumptions on how the count is performed:

**C1)** All possible maps $\alpha : \{0,1\}^k \to \{0,1\}^k$ can be learned by the neural model, that is $\mathcal{A}$ is complete and of cardinality $|\mathcal{A}| = \left(2^k\right)^{2^k}$

**C2)** The support of $p(\mathbf{c}^*)$ is complete, that is $\mathrm{supp}(\mathbf{c}^*) = 2^k$

Here, assumption (**C1**) allows to count the total number of maps by considering the limit case where each $\alpha(\mathbf{c})$ can be predicted independently from $\alpha(\mathbf{c}')$, for $\mathbf{c} \neq \mathbf{c}'$. It is possible to derive the total count of possible optimal $\alpha$'s [20], given by:

$$\#\text{opt-}\alpha\text{'s}(\mathbf{C1}, \mathbf{C2}) = \prod_{\mathbf{c} \in \{0,1\}^k} |\mathcal{E}(\mathbf{c}, \mathsf{K})|^{|\mathcal{E}(\mathbf{c}, \mathsf{K})|} \tag{8}$$

where the set $\mathcal{E}(\mathbf{c}, \mathsf{K}) \subset \{0,1\}^k$ contains all concepts C that yield the same result under $\beta_{\mathsf{K}}$, and it is defined as $\mathcal{E}(\mathbf{c}, \mathsf{K}) := \{\mathbf{c}' : \beta_{\mathsf{K}}(\mathbf{c}') = \beta_{\mathsf{K}}(\mathbf{c})\}$. In other terms, $\mathcal{E}(\mathbf{c}, \mathsf{K}) \subset \{0,1\}^k$ is the equivalence class for $\mathbf{c}$ and logic K. It is possible to relax **C2** by capturing a more general case and showing that RSs can be even more. The underlying idea is that for all $\mathbf{c}'' \notin \mathrm{supp}(\mathbf{c}^*)$, *i.e.*, not appearing in the training set, a map $\alpha$ can predict whatever element in $\{0,1\}^k$ without affecting the training likelihood. This gives an even bigger number of solutions provided by:

$$\#\text{opt-}\alpha\text{'s}(\mathbf{C1}) = \prod_{\mathbf{c} \in \mathrm{supp}(\mathbf{C}^*)} |\mathcal{E}(\mathbf{c}, \mathsf{K})|^{|\mathcal{E}(\mathbf{c}, \mathsf{K})|} \prod_{\mathbf{c} \notin \mathrm{supp}(\mathbf{C}^*)} 2^k \tag{9}$$

However, relaxing **C1** makes finding an explicit expression for counting optimal solutions more complicated. We then resort to formal methods to show that this can be done algorithmically.

**Algorithmic implementation with `countrss`.** We detail the encoding of maps $\alpha$ and their counting in propositional logic, under the previous assumptions **A1** and **A2**. We denote with $\mathbb{B} = \{0, 1\}$ and matrix element $\mathbf{L}_{i,j}$ as $\mathbf{L}[i, j]$. For ease of exposition, we additionally assume that all concepts $c_i$, for $i \in \{1, \ldots, k\}$, have the same number of values $b \in \mathbb{N}^+$.

We introduce $\mathbf{O} \in \mathbb{B}^{k \times k}$ the boolean matrix encoding the mapping between the ground-truth concepts $\mathbf{c}^*$ and the predicted concepts $\mathbf{c}$, where $k \in \mathbb{N}^+$ is the number of concepts. Intuitively, $\mathbf{O}$ determines what learned concepts depend on the ground truth ones. For example, it may be the case that in `Kand-Logic`, the concept $c_{\texttt{shape}}$ depends on $c^*_{\texttt{color}}$ and $c_{\texttt{color}}$ depends on $c^*_{\texttt{shape}}$.

The full mapping from ground truth to predicted concept values is defined by the block matrix $\mathbf{A} \in \mathbb{B}^{(k \cdot b) \times (k \cdot b)}$. This mapping encodes how the values of the ground truth concepts are mapped to the values of the predicted predicted. For example, in `Kand-Logic` it may be the case that $\alpha : red \mapsto \square$ and $\alpha : \square \mapsto red$, and so on for other colors and shapes. We require that this assignment of $\mathbf{A}$ is consistent with the assignment of $\mathbf{O}$. This means that the $i, j$-block of $\mathbf{A}$ has non-zero entries if and only if the $i$-th ground truth concept is mapped onto the $j$-th predicted concept:

$$\bigwedge_{i=1}^{k} \bigwedge_{j=1}^{k} \left[ \mathbf{O}[i,j] \Leftrightarrow \left( \bigvee_{x=ib}^{i(b+1)-1} \bigvee_{y=jb}^{j(b+1)-1} \mathbf{A}[x,y] \right) \right]. \tag{10}$$

$\mathbf{A}$ also must have exactly one positive entry for each column, encoding the fact that a single ground-truth value cannot be mapped to two or more values by the model. Again, multiple ground-truth values can still end up collapsing into the same predicted value:

$$\bigwedge_{j=1}^{k \cdot b} \mathsf{OHE}(\mathbf{A}[1,j], ..., \mathbf{A}[k \cdot b, j]). \tag{11}$$

Out of every assignment to $\mathbf{A}$ satisfying the constraints above, only those that are consistent with the supervision can achieve optimal predictive performance and therefore be potential RSs. Let $\mathbf{c}^* \in \mathbb{B}^{k \cdot b}$ denote the boolean vector encoding the ground truth concept values appearing in the support of $p(\mathbf{c}^*)$, and let $\hat{\mathbf{c}} \in \mathbb{B}^{k \cdot b}$ denote the predicted concepts for the ground-truth concept, which is defined as the boolean dot product (denoted as $\otimes$) of $\mathbf{c}^*$ with $\mathbf{A}$:

$$\bigwedge_{\mathbf{c}^* \in \mathsf{supp}(\mathbf{c}^*)} (\hat{\mathbf{c}} \Leftrightarrow \mathbf{A} \otimes \mathbf{c}^*). \tag{12}$$

Finally, by denoting with $\mathsf{K}$ and $\mathbf{y}^*$ the logical encoding of the task and the ground-truth label for the $\mathbf{c}^*$ (corresponding to $\mathbf{y}^* = \beta_{\mathsf{K}}(\mathbf{c}^*)$) respectively, we constrain the model predictions to be correct, which in turn forces the values of $\mathbf{A}$ to comply with condition (**D1**):

$$\bigwedge_{\mathbf{c}^* \in \mathsf{supp}(\mathbf{c}^*)} (\mathsf{K}(\mathbf{c}_d) \Leftrightarrow \mathbf{y}^*). \tag{13}$$

The full encoding is the conjunction of Eq. (10), Eq. (11), Eq. (12), and Eq. (13). This is fed to a model counter, whose output equals the number of possible assignments to $\mathbf{A}$ that satisfy the formula. When this number is above 1, RSs are present in the learning problem. This number indicates the optimal maps $\alpha$'s (according to the specifics of the constraints) that can be learned by the NeSy model.

Notice that, without any additional constraint on $\mathbf{O}$, the count would enumerate all RSs as done in Eq. (9). We instead search for more specific RSs that relax condition **C1**. In our setup, we constrain each ground-truth concept $c_i^*$ to be mapped to a single extracted concept $c_i$. This condition is typically referred to as *completeness* [102], that is there are no copies or repetitions of ground-truth concepts in the learned ones. For example, we avoid counting solutions where the map $\alpha$ predicts two concepts, say $c_1$ and $c_2$, only depending on one ground-truth concept, say $c_1^*$. Notice that, however, multiple concepts $c_i^*$ can still affect a single concept in $c_j$. Formally, we achieve this by enforcing exactly one (OHE) positive value in each column of $\mathbf{O}$:

$$\bigwedge_{j=1}^{k} \mathsf{OHE}(\mathbf{O}[1,j], ..., \mathbf{O}[k,j]) \tag{14}$$

where $\mathsf{OHE}(a_1, \ldots, a_k)$ is one *if and only if* there is only one $a_i = 1$ and the remaining are zero, otherwise zero. Basically, the matrix $\mathbf{O}$ encodes all maps of concept indices, a necessary element whenever there is no clear notion of ordering among concepts (CPLX $\mathbf{x}$).

Then, this additional constraint can be added to the previous conjunction and used to evaluate the number of RSs. One perk of `countrss` is that many additional constraints can be added to the conjunction of Eq. (10), Eq. (11), Eq. (12), and Eq. (13) and evaluate different situations depending on the expected model architectural biases. By adding

$$\bigwedge_{j=1}^{k} \mathsf{OHE}(\mathbf{O}[j, 1], ..., \mathbf{O}[j, k]) \tag{15}$$

to Eq. (14), we then require that only permutation maps are considered, that is one $\mathbf{c}_i$ only depends on one $\mathbf{c}_{\pi(i)}^*$, where $\pi$ is a permutation of $k$ elements. Further constraints can also be added to include concept supervision.

## B  Experiments: Additional Details

All experiments were conducted using Python 3.8 and PyTorch 1.13, executed on a single A5000 GPU. The implementation of DPL was taken from [25]. For LTN, NN, CBM, and CLIP, new implementations were developed from scratch. LTN models were developed employing LTNtorch [103]. As for CLIP, we leveraged the implementation of [104]. For all the datasets, we used the pre-trained (with contrastive learning for image-caption matching, see [56]) visual transformer ViT-B/32 to this end, passing all input images to first a rescaling transformation of the image to $224 \times 224 \times 3$ to comply with the backbone layer. Visual embeddings are then saved and made available in our supplementary material for successive fine-tuning of neural predictors.

For the dataset generation process of `rsbench`, we utilized Python 3.7 along with Blender 2.91, leveraging the `bpy` Python package set to version 2.91a0. The methodology for generating CLE4EVR variants and SDD-OIA was inspired by the work of Johnson et al. [65]. The images, which require Blender rendering, were generated using a Tesla K40c GPU.

SDD-OIA was generated using seed 0, with random splits of 0.7 for train, 0.15 for validation, and 0.15 for test. The configuration was 0.9 in-distribution and 0.1 out-of-distribution. The dataset comprises 6820 training examples, 1464 validation examples, 1464 test examples, and 1000 OOD examples. Kand-Logic and MNAdd-EvenOdd are the same datasets used in [25]. MNMath was generated using seed 0 with random splits, maintaining an $80/20$ ratio for in-distribution and out-of-distribution data. It contains $1,000$ training samples, 200 validation samples, 300 test samples, and 200 OOD samples. MNLogic was also generated using seed 0 and random splits, with an $80/20$ ID/OOD ratio. It consists of $1,000$ training samples, 200 validation samples, 300 test samples, and 300 OOD samples.

All models were trained using end-to-end training, providing supervision on the ground truth labels and not on the concept labels, except for CBM where few concept supervision has been provided.

### B.1  Additional experiments: MNMath and MNLogic

In this section, we discuss additional experiments conducted on MNMath and MNLogic datasets.

For MNMath, we chose a task involving two equations with 8 digits (4 per equation), where $\mathbf{x} = (\mathbf{x}_1, \mathbf{x}_2, \mathbf{x}_3, \mathbf{x}_4, \mathbf{x}_5, \mathbf{x}_6, \mathbf{x}_7, \mathbf{x}_8)$. We designed a multi-task setup: one task predicts whether $y_1 = \mathbb{1}\{\mathbf{x}_1 + \mathbf{x}_2 = \mathbf{x}_3 + \mathbf{x}_4\}$, and the other checks if $y_2 = \mathbb{1}\{\mathbf{x}_5 \cdot \mathbf{x}_6 = \mathbf{x}_7 \cdot \mathbf{x}_8\}$ holds.

This version of MNMath includes all possible digit values (0 to 9). We chose this setup because it is particularly prone to reasoning shortcuts. Specifically, if all digits are mapped to a fixed value, *e.g.*, $\boxed{0}, \ldots, \boxed{9} \to 4$, they solve the MNMath task.

We evaluated three models (DPL, NN, and CBM) on this benchmark. The setup mirrors previous experiments (Section 4), where each model receives supervision only on the final labels so that the concepts are treated as latent variables. As CBM require concept supervision, we gave supervision on few concepts, specifically $0, 5$ and $9$. In this case the model predicts the concept (the digit value) for each digit independently. Following that, the model produces two labels, representing whether the two formulas are true, respectively.

The results in Table 6 summarize the performance of all models, averaged across five different seeds. The table highlights that all models struggle with reasoning shortcuts (RSs), reflected in the low concept accuracy ($\text{Acc}_C$) and concept fidelity ($\text{F}_1(C)$) scores, particularly for `NN` and `CBM`. However, despite these issues, the models still manage to achieve moderate to high performance on label prediction metrics ($\text{Acc}_Y$ and $\text{F}_1(Y)$).

Table 6: Results on `MNMath`

|  | $\text{Acc}_Y(\uparrow)$ | $\text{F}_1(Y)(\uparrow)$ | $\text{Acc}_C(\uparrow)$ | $\text{F}_1(C)(\uparrow)$ | $\text{Cls}(C)(\downarrow)$ |
|---|---|---|---|---|---|
| DPL | $0.80 \pm 0.10$ | $0.73 \pm 0.13$ | $0.11 \pm 0.01$ | $0.03 \pm 0.02$ | $0.01 \pm 0.01$ |
| CBM$^\dagger$ | $0.75 \pm 0.01$ | $0.67 \pm 0.01$ | $0.22 \pm 0.04$ | $0.11 \pm 0.03$ | $0.68 \pm 0.15$ |
| NN$^*$ | $0.75 \pm 0.01$ | $0.67 \pm 0.01$ | $0.10 \pm 0.01$ | $0.03 \pm 0.01$ | $0.80 \pm 0.11$ |

For `MNLogic`, we focused on evaluating the XOR operation on 4 bits. In this case, the input is $\mathbf{x} = (\mathbf{x}_1, \mathbf{x}_2, \mathbf{x}_3, \mathbf{x}_4)$, and the task is to compute the output $y = \mathbf{x}_1 \oplus \mathbf{x}_2 \oplus \mathbf{x}_3 \oplus \mathbf{x}_4$.

We chose the XOR operation for its inherent ambiguity, which can lead to reasoning shortcuts, as discussed in previous work [20].

We evaluated the `DPL`, `NN`, and `CBM` models under the same setup as in `MNMath`, with the sole exception that there is no weight sharing among the components processing each digit, which makes the task more challenging. Unlike in `MNMath`, the `CBM` model did not receive concept supervision, as supervision for either 0 or 1 would suffice for learning the concept correctly.

The results in Table 7 reflect averages across five seeds. While all models demonstrated high performance on the task (as indicated by $\text{Acc}_Y$ and $\text{F}_1(Y)$), they exploited unintended concepts, evident from the $\text{Acc}_C$ and $\text{F}_1(C)$ metrics. Interestingly, the 50% concept accuracy achieved by all models indicates that there is no inherent preference for any model to favor one solution over another, whether it be the identity or the reasoning shortcut ( illustrated in Fig. 3).

Table 7: Results on `MNLogic`

|  | $\text{Acc}_Y(\uparrow)$ | $\text{F}_1(Y)(\uparrow)$ | $\text{Acc}_C(\uparrow)$ | $\text{F}_1(C)(\uparrow)$ | $\text{Cls}(C)(\downarrow)$ |
|---|---|---|---|---|---|
| DPL | $0.99 \pm 0.01$ | $0.99 \pm 0.01$ | $0.51 \pm 0.06$ | $0.47 \pm 0.05$ | $0.01 \pm 0.01$ |
| CBM$^\dagger$ | $0.95 \pm 0.10$ | $0.89 \pm 0.23$ | $0.50 \pm 0.05$ | $0.48 \pm 0.05$ | $0.01 \pm 0.01$ |
| NN$^*$ | $0.98 \pm 0.01$ | $0.60 \pm 0.20$ | $0.46 \pm 0.05$ | $0.41 \pm 0.10$ | $0.10 \pm 0.20$ |

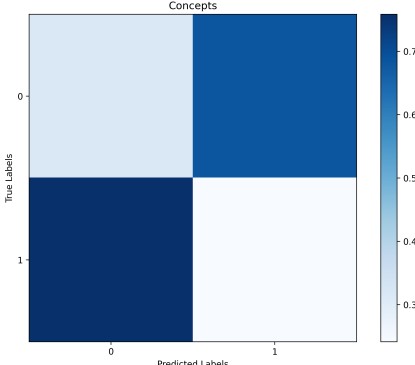

Figure 3: `MNLogic` reasoning shortcut

## B.2 Loss Functions

For `NN`, `CBM`, `DPL` and `CLIP`, the loss function corresponding to log-likelihood maximization is given by the cross-entropy loss, defined as:

$$\mathcal{L}_{\text{CE}}(\mathbf{x}) = -\sum_{i=1}^{\ell} y_i \log(\hat{y}_i) \tag{16}$$

where $y_i = 1$ if $i$ is the ground-truth label for input $\mathbf{x}$, and $\hat{y}_i$ is the predicted probability for class $i$ by the model when passed the input example $\mathbf{x}$.

LTN is the only one that differs and utilizes the Logic Tensor Network loss [59]. In LTN, the neural networks represent First-Order Logic predicates (*e.g.*, the predicate $\mathrm{Digit}$ in the MNAdd task). Formulas are constructed recursively by using fuzzy logic operators (*e.g.*, fuzzy quantifiers and logical connectives). The LTN loss imposes on learning the parameters of such predicates in such a way the satisfaction of the knowledge base is maximized. Given a First-Order Logic knowledge base K, the loss can be defined as:

$$\mathcal{L}_{\text{ltn}}(\mathbf{x}) = 1 - \text{SatAgg}_{\phi \in \mathsf{K}} \, \mathcal{G}_\theta(\mathbf{x}; \phi) \tag{17}$$

where $\phi$ is a formula contained in K (like the addition for MNAdd), $\text{SatAgg}$ is an aggregator function that measures overall satisfaction of K, and $\mathcal{G}_\theta(\mathbf{x}; \phi)$ is the LTN-*grounding* (i.e., evaluation) of $\phi$ given $\mathbf{x}$. This special operator is meant to map symbols from the logical domain (*e.g.*, the symbolic representation of the addition with "+") to the real domain (*e.g.*, its computational graph performing the addition). Hence, $\mathcal{G}_\theta(\mathbf{x}; \phi)$ can be seen as a fuzzy truth value resulting from the evaluation of logical formula $\phi$ on input $\mathbf{x}$. $\theta$ are the parameters of the learnable predicates contained in $\phi$. In the case of multi-label predictions, parameters $\theta$ of the neural network are shared among different formulas $\phi \in \mathsf{K}$ (*e.g.*, like in MNMath where different equations appear in the system). Learning is then performed by differentiating the above expression.

The value of the loss depends on the semantics of the fuzzy logic operators used to approximate each logical connective (*e.g.*, AND, OR, IMPLICATION) and quantifier (*e.g.*, EXISTS, FORALL).

For CBM, we provided partial supervision by selecting only a few concept classes for supervision, rather than supervising all concepts. The concept supervision was also implemented using cross-entropy loss, specifically applied to the concepts. The cross-entropy loss for concepts is given by:

$$\mathcal{L}_{\text{concept}}(\mathbf{x}) = -\frac{1}{k} \sum_{i=1}^{k} m_i \sum_{b=1}^{B_i} c_{ib} \log(\hat{c}_{ib}) \tag{18}$$

where $c_{ib} = 1$ if $i$-th ground-truth concept has value $b$ for input $\mathbf{x}$, and $\hat{c}_{ib}$ is the predicted probability for concept $i$ with value $b$. Here, $B_i$ denotes the cardinality of the $i$-th concept, and $m_i = 1$ if the concept $i$ is supervised, otherwise being zero.

**Entropy.** To further explore the concept space, for LTN and DPL in MNAdd-EvenOdd, we applied an entropy loss on the bottleneck of the concepts. The entropy loss encourages diversity in the concept representations and is defined as:

$$\mathcal{L}_{\text{entropy}} = -\sum_{i=1}^{k} \frac{\hat{c}_i \log(\hat{c}_i)}{\log(k)} \tag{19}$$

where $\hat{c}_i$ is the average probability, evaluated over the batch elements.

**Combined losses.** In scenarios where both ground truth labels and concept labels were used, the total loss is a weighted sum of the cross-entropy loss and the concept loss:

$$\mathcal{L}_{\text{total}}(\mathbf{x}) = \mathcal{L}_{\text{CE}}(\mathbf{x}) + w_c \mathcal{L}_{\text{concept}}(\mathbf{x}) + w_h \mathcal{L}_{\text{entropy}} \tag{20}$$

where $w_c$ and $w_h$ are hyperparameters that controls the trade-off between the two losses. For LTN the equivalent can be obtained by substituting $\mathcal{L}_{\text{CE}}(\mathbf{x})$ with $\mathcal{L}_{\text{ltn}}(\mathbf{x})$.

## B.3 Model Selection

All experiments rely on the Adam optimizer [105]. Hyperparameters were selected through a comprehensive grid search over predefined ranges, considering the macro $f_1$ performance metric on a validation set. All experiments were run for 40 epochs employing early stopping, by saving the model which performs best in f1 score on the validation set. The experiments, aside from MNMath and MNLogic, were conducted using 10 different seeds: 123, 456, 789, 1011, 1213, 1415, 1617, 1819, 2021, and 2122. In contrast, MNMath and MNLogic were tested across 5 different seeds: 1415, 1617, 1819, 2021, and 2223.

For all experiments, the learning rate $\gamma$ was fine-tuned within the range of $10^{-4}$ to $10^{-2}$. We found that the best performing models tended to have learning rates around $10^{-3}$, striking a balance between convergence speed and stability.

The batch size $\nu$ varied between 32 and 512, while the weight decay $\omega$ spanned from $10^{-4}$ to 0. We observed that smaller batch sizes generally resulted in more stable training dynamics, particularly for complex models, while moderate weight decay values helped prevent overfitting.

For CBM, since it requires concept supervision, we tuned the weight of the concept supervision $w_c$ among 1, 2, and 5.

When entropy was required to make the model converge, we tuned the weight of the entropy $w_h$ loss among 0.2, 0.5, 0.8, 1, and 2.

Additionally, for LTN, the hyper-parameter p for quantifiers was adjusted within the range of 2 to 10 with a step size of 2. Moreover, we tuned different fuzzy logic semantics for the fuzzy operators, specifically for AND, OR and IMPLICATION, such as Gödel, Product and ukasiewicz for AND and OR, while Gödel, Product, ukasiewicz, Goguen and Kleene-Dienes for IMPLICATION.

We set the exponential decay rate $\beta$ to 0.99 for all experiments, as we empirically observed that it provides the best performance for our tasks.

Below, you find all the hyperparameters which performed the best on our datasets.

**Hyperparameters for SDD-OIA:**

- DPL, $\gamma = 10^{-2}$, $\nu = 128$, and $\omega = 10^{-4}$;
- LTN, $\gamma = 10^{-3}$, $\nu = 32$, $\omega = 0$, and p $= 2$, AND, OR, IMPLICATION set to Product;
- NN, $\gamma = 10^{-3}$, $\nu = 32$, and $\omega = 10^{-4}$;
- CBM, $\gamma = 10^{-2}$, $\nu = 512$, $\omega = 10^{-4}$, and $w_c = 2$;
- CLIP, $\gamma = 10^{-3}$, $\nu = 32$, and $\omega = 10^{-2}$.

**Hyperparameters for Kand-Logic:**

- DPL, $\gamma = 10^{-4}$, $\nu = 32$, and $\omega = 0$;
- LTN, $\gamma = 10^{-3}$, $\nu = 128$, $\omega = 10^{-3}$, p $= 8$, and $w_h = 0.8$ AND set to Godel, OR and IMPLICATION set to Product;
- NN, $\gamma = 10^{-3}$, $\nu = 256$, and $\omega = 10^{-1}$;
- CBM, $\gamma = 10^{-4}$, $\nu = 128$, $\omega = 10^{-2}$, and $w_c = 2$;
- CLIP, $\gamma = 10^{-3}$, $\nu = 256$, and $\omega = 10^{-1}$.

**Hyperparameters for MNAdd-EvenOdd:**

- DPL, $\gamma = 10^{-3}$, $\nu = 32$, $\omega = 10^{-4}$ and $w_h = 1$;
- LTN, $\gamma = 10^{-3}$, $\nu = 64$, $\omega = 10^{-4}$, p $= 6$, $w_h = 10$, AND set to Godel, OR and IMPLICATION set to Product;
- NN, $\gamma = 10^{-3}$, $\nu = 32$, and $\omega = 10^{-1}$;
- CBM, $\gamma = 10^{-3}$, $\nu = 32$, $\omega = 0$, and $w_c = 2$;
- CLIP, $\gamma = 10^{-2}$, $\nu = 128$, and $\omega = 10^{-2}$.

**Hyperparameters for MNMath:**

- DPL, $\gamma = 10^{-3}$, $\nu = 64$, $\omega = 10^{-4}$ and $w_h = 0$;
- NN, $\gamma = 10^{-4}$, $\nu = 64$, and $\omega = 10^{-4}$;
- CBM, $\gamma = 10^{-3}$, $\nu = 64$, $\omega = 10^{-4}$, and $w_c = 1$.

**Hyperparameters for MNLogic:**

- DPL, $\gamma = 10^{-3}$, $\nu = 64$, $\omega = 10^{-4}$ and $w_h = 0$;
- NN, $\gamma = 10^{-3}$, $\nu = 64$, and $\omega = 10^{-4}$;
- CBM, $\gamma = 10^{-4}$, $\nu = 64$, $\omega = 10^{-4}$, and $w_c = 1$;

## B.4 Model Architectures

`SDD-OIA`:  For `SDD-OIA`, concerning DPL, LTN and CBM, we adopted a pretrained ResNet-18 [106] on ImageNet [107] as concept extractor as outlined in Table 8 and Table 10. In the tables, BasicBlock consists of two convolutional layers with batch normalization and ReLU activation, followed by a residual connection. Instead, we defined a convolutional architecture for `NN` as shown in Table 9. While for processing `CLIP` embeddings we defined a multi-layer perceptron as depicted in Table 11.

`Kand-Logic`:  As motivated in [25], for LTN, DPL, and CBM, we adopted a preprocessed version of the dataset with rescaled objects extracted via bounding boxes. In this scenario, each dataset example contains nine objects, with three objects per figure, ordered by their distance from the figure's origin, and the network processes one object at a time. For `CLIP` and `NN`, we employed the original dataset, where the network processes the entire example at once. As far as the architectures are concerned, we employed a convolutional neural network, specifically Table 16 for DPL and LTN,  Table 18 for CBM and  Table 17 for `NN`, respectively. Conversely, for handling `CLIP` embeddings, we implemented a multi-layer perceptron, as detailed in Table 19.

`MNAdd-EvenOdd`:  In the case of `MNAdd-EvenOdd`, we utilized a convolutional neural network, with architectures specified in Table 12 for DPL and LTN, in Table 13 for CBM, and in Table 14 for `NN`. For processing `CLIP` embeddings, a multi-layer perceptron was employed, as described in Table 15.

`MNMath`:  For `MNMath`, we used the same architectures as in `MNAdd-EvenOdd`, specifically a convolutional neural network. The architectures for DPL are detailed in Table 12, while those for CBM are listed in Table 13. For `NN`, we utilized the architecture described in Table 22.

`MNLogic`:  For `MNLogic`, we also applied convolutional networks. The architectures for both DPL and CBM are provided in Table 21, and the `NN` architecture is outlined in Table 20.

All the architectures process each digit individually, except for `CLIP` and `NN`. `CLIP` takes the full image embeddings as input.

Table 8: DPL and LTN architecture for `SDD-OIA`

| INPUT | LAYER TYPE | PARAMETER | ACTIVATION |
|---|---|---|---|
| $(3, 387, 469)$ | Convolution | depth=64, kernel=7, stride=2, padding=3 | |
| $(64, 194, 235)$ | BatchNorm2d | dim=64 | ReLU |
| $(64, 194, 235)$ | MaxPool2d | kernel=3, stride=2, padding=1 | ReLU |
| $(64, 97, 118)$ | 2xBasicBlock | depth=64, kernel=(1, 1), stride=(1, 1), padding=(1, 1) | |
| $(64, 97, 118)$ | 2xBasicBlock | depth=128, kernel=(1, 1), stride=(1, 1), padding=(1, 1) | |
| $(128, 49, 59)$ | 2xBasicBlock | depth=256, kernel=(1, 1), stride=(1, 1), padding=(1, 1) | |
| $(256, 25, 30)$ | 2xBasicBlock | depth=512, kernel=(1, 1), stride=(1, 1), padding=(1, 1) | |
| $(512, 13, 15)$ | AvgPool2d | dim=(1, 1) | |
| $(512, 1, 1)$ | Flatten | dim=512 | |

Table 9: `NN` architecture for `SDD-OIA`

| INPUT | LAYER TYPE | PARAMETER | ACTIVATION |
|---|---|---|---|
| $(3, 387, 469)$ | Convolution | depth=16, kernel=3, stride=1, padding=1 | ReLU |
| $(16, 387, 469)$ | MaxPool2d | kernel=2 | |
| $(16, 193, 234)$ | Convolution | depth=32, kernel=3, stride=1, padding=1 | ReLU |
| $(32, 193, 234)$ | MaxPool2d | kernel=2 | |
| $(32, 96, 117)$ | Convolution | depth=64, kernel=3, stride=1, padding=1 | |
| $(64, 96, 117)$ | MaxPool2d | kernel=2 | |
| $(64, 48, 58)$ | Convolution | depth=128, kernel=3, stride=1, padding=1 | |
| $(128, 48, 58)$ | MaxPool2d | kernel=2 | |
| $(128, 24, 29)$ | Convolution | depth=256, kernel=3, stride=1, padding=1 | |
| $(256, 24, 29)$ | MaxPool2d | kernel=2 | |
| $(256, 12, 14)$ | Convolution | depth=512, kernel=3, stride=1, padding=1 | |
| $(512, 12, 14)$ | MaxPool2d | kernel=2 | |
| $(512, 6, 7)$ | Flatten | dim=21504 | |
| $(21504)$ | Linear | dim=128 | ReLU |
| $(128)$ | Linear | dim=64 | ReLU |
| $(64)$ | Linear | dim=4 | Sigmoid |

Table 10: CBM architecture for SDD-OIA

| INPUT | LAYER TYPE | PARAMETER | ACTIVATION |
|---|---|---|---|
| (3, 387, 469) | Convolution | depth=64, kernel=7, stride=2, padding=3 | |
| (64, 194, 235) | BatchNorm2d | dim=64 | ReLU |
| (64, 194, 235) | MaxPool2d | kernel=3, stride=2, padding=1 | ReLU |
| (64, 97, 118) | 2xBasicBlock | depth=64, kernel=(1, 1), stride=(1, 1), padding=(1, 1) | |
| (64, 97, 118) | 2xBasicBlock | depth=128, kernel=(1, 1), stride=(1, 1), padding=(1, 1) | |
| (128, 49, 59) | 2xBasicBlock | depth=256, kernel=(1, 1), stride=(1, 1), padding=(1, 1) | |
| (256, 25, 30) | 2xBasicBlock | depth=512, kernel=(1, 1), stride=(1, 1), padding=(1, 1) | |
| (512, 13, 15) | AvgPool2d | dim=(1, 1) | |
| (512, 1, 1) | Flatten | dim=512 | |
| (512) | Linear | dim=4, bias=True | Sigmoid |

Table 11: CLIP architecture for SDD-OIA

| INPUT | LAYER TYPE | PARAMETER | ACTIVATION |
|---|---|---|---|
| (512, 1) | Linear | dim=128, bias=True | ReLU |
| (128) | Linear | dim=64, bias=True | ReLU |
| (64) | Linear | dim=4, bias=True | Sigmoid |

Table 12: DPL and LTN architecture for MNAdd-EvenOdd

| INPUT | LAYER TYPE | PARAMETER | ACTIVATION |
|---|---|---|---|
| (1, 28, 56) | Convolution | depth=32, kernel=4, stride=2, padding=1 | ReLU |
| (32, 14, 28) | Dropout | $p = 0.5$ | |
| (32, 14, 28) | Convolution | depth=64, kernel=4, stride=2, padding=1 | ReLU |
| (64, 7, 14) | Dropout | $p = 0.5$ | |
| (64, 7, 14) | Convolution | depth=128, kernel=4, stride=2, padding=1 | ReLU |
| (128, 3, 7) | Flatten | | |
| (2688) | Linear | dim=20, bias = True | |

Table 13: CBM architecture for MNAdd-EvenOdd

| INPUT | LAYER TYPE | PARAMETER | ACTIVATION |
|---|---|---|---|
| (1, 28, 56) | Convolution | depth=32, kernel=4, stride=2, padding=1 | ReLU |
| (32, 14, 28) | Dropout | $p = 0.5$ | |
| (32, 14, 28) | Convolution | depth=64, kernel=4, stride=2, padding=1 | ReLU |
| (64, 7, 14) | Dropout | $p = 0.5$ | |
| (64, 7, 14) | Convolution | depth=128, kernel=4, stride=2, padding=1 | ReLU |
| (128, 3, 7) | Flatten | | |
| (2688) | Linear | dim=20, bias = True | ReLU |
| (20) | Linear | dim=19, bias = True | |

Table 14: NN architecture for MNAdd-EvenOdd

| INPUT | LAYER TYPE | PARAMETER | ACTIVATION |
|---|---|---|---|
| (1, 28, 56) | Convolution | depth=16, kernel=3, stride=1, padding=1 | ReLU |
| (16, 28, 56) | MaxPool2d | kernel=2, stride=2 | |
| (16, 14, 28) | Convolution | depth=32, kernel=3, stride=1, padding=1 | ReLU |
| (32, 14, 28) | MaxPool2d | kernel=2, stride=2 | |
| (32, 7, 14) | Flatten | | |
| (3136) | Linear | dim=128, bias=True | ReLU |
| (128) | Linear | dim=64, bias=True | ReLU |
| (64) | Linear | dim=19, bias=True | |

Table 15: CLIP architecture for MNAdd-EvenOdd

| INPUT | LAYER TYPE | PARAMETER | ACTIVATION |
|---|---|---|---|
| (1024, 1) | Linear | dim=256, bias=True | ReLU |
| (256) | Linear | dim=64, bias=True | ReLU |
| (64) | Linear | dim=19, bias=True | |

Table 16: DPL and LTN architecture for Kand-Logic

| INPUT | LAYER TYPE | PARAMETER | ACTIVATION |
|---|---|---|---|
| $(3, 28, 28)$ | Flatten | | |
| $(2352)$ | Linear | dim=256, bias=True | ReLU |
| $(256)$ | Linear | dim=128, bias=True | ReLU |
| $(128)$ | Linear | dim=8, bias = True | |

Table 17: NN architecture for Kand-Logic

| INPUT | LAYER TYPE | PARAMETER | ACTIVATION |
|---|---|---|---|
| $(3, 64, 192)$ | Convolution | depth=16, kernel=3, stride=1, padding=1 | ReLU |
| $(16, 64, 192)$ | MaxPool2d | kernel=2, stride=2 | |
| $(16, 32, 96)$ | Convolution | depth=32, kernel=3, stride=1, padding=1 | ReLU |
| $(32, 32, 96)$ | MaxPool2d | kernel=2, stride=2 | |
| $(32, 16, 48)$ | Convolution | depth=64, kernel=3, stride=1, padding=1 | ReLU |
| $(64, 16, 48)$ | MaxPool2d | kernel=2, stride=2 | |
| $(64, 8, 24)$ | Convolution | depth=128, kernel=3, stride=1, padding=1 | ReLU |
| $(128, 8, 24)$ | MaxPool2d | kernel=2, stride=2 | |
| $(128, 4, 12)$ | Convolution | depth=256, kernel=3, stride=1, padding=1 | ReLU |
| $(256, 4, 12)$ | MaxPool2d | kernel=2, stride=2 | |
| $(256, 2, 6)$ | Flatten | | |
| $(3072)$ | Linear | dim=512, bias=True | ReLU |
| $(512)$ | Linear | dim=64, bias=True | ReLU |
| $(64)$ | Linear | dim=1, bias=True | |

Table 18: CBM architecture for Kand-Logic

| INPUT | LAYER TYPE | PARAMETER | ACTIVATION |
|---|---|---|---|
| $(3, 28, 28)$ | Flatten | | |
| $(2352)$ | Linear | dim=256, bias=True | ReLU |
| $(256)$ | Linear | dim=128, bias=True | ReLU |
| $(128)$ | Linear | dim=8, bias = True | |
| $(8)$ | Linear | dim=6, bias = True | |
| $(6)$ | Linear | dim=2, bias = True | |

Table 19: CLIP architecture for Kand-Logic

| INPUT | LAYER TYPE | PARAMETER | ACTIVATION |
|---|---|---|---|
| $(1536, 1)$ | Linear | dim=256, bias=True | ReLU |
| $(256)$ | Linear | dim=64, bias=True | ReLU |
| $(64)$ | Linear | dim=1, bias=True | Sigmoid |

Table 20: NN architecture for MNLogic

| INPUT | LAYER TYPE | PARAMETER | ACTIVATION |
|---|---|---|---|
| $(1, 28, 112)$ | Convolution | depth=16, kernel=3, padding=1 | ReLU |
| $(16, 28, 112)$ | MaxPool2d | kernel=2, stride=2 | |
| $(16, 14, 56)$ | Convolution | depth=32, kernel=3, padding=1 | ReLU |
| $(32, 14, 56)$ | MaxPool2d | kernel=2, stride=2 | |
| $(32, 7, 28)$ | Convolution | depth=64, kernel=3, padding=1 | ReLU |
| $(64, 7, 28)$ | Flatten | | |
| $(12544)$ | Linear | dim=128, bias=True | ReLU |
| $(128)$ | Linear | dim=64, bias=True | ReLU |
| $(64)$ | Linear | dim=2, bias=True | |

Table 21: CBM and DPL architecture for MNLogic

| INPUT | LAYER TYPE | PARAMETER | ACTIVATION |
|---|---|---|---|
| $(1, 28, 28)$ | Convolution | depth=32, kernel=3, padding=1, stride=1 | ReLU |
| $(32, 28, 28)$ | MaxPool2d | kernel=2, stride=2 | |
| $(32, 14, 14)$ | Convolution | depth=64, kernel=3, padding=1, stride=1 | ReLU |
| $(64, 14, 14)$ | MaxPool2d | kernel=2, stride=2 | |
| $(64, 7, 7)$ | Flatten | | |
| $(3136)$ | Linear | dim=128, bias=True | ReLU |
| $(128)$ | Linear | dim=2, bias=True | |

Table 22: NN architecture for `MNMath`

| INPUT | LAYER TYPE | PARAMETER | ACTIVATION |
|---|---|---|---|
| $(1, 28, 224)$ | Convolution | depth=16, kernel=3, padding=1 | ReLU |
| $(16, 28, 224)$ | MaxPool2d | kernel=2, stride=2 | |
| $(16, 14, 112)$ | Convolution | depth=32, kernel=3, padding=1 | ReLU |
| $(32, 14, 112)$ | MaxPool2d | kernel=2, stride=2 | |
| $(32, 7, 56)$ | Convolution | depth=64, kernel=3, padding=1 | ReLU |
| $(64, 7, 56)$ | Flatten | | |
| $(25088)$ | Linear | dim=128, bias=True | ReLU |
| $(128)$ | Linear | dim=64, bias=True | ReLU |
| $(64)$ | Linear | dim=2, bias=True | |

# C   Code, Data Sets and Generators

In the following, we discuss: 1) code and data licensing Appendix C.1, 2) how the data was collected and organised Appendix C.4, 3) what kind of information it contains Appendix C.5, 4) how it should be used ethically and responsibly Appendix C.2, 5) how it will be made available and maintained Appendix C.3. All data, generators, metadata, and experimental code for reproducing the results are available at: `https://unitn-sml.github.io/rsbench`.

Detailed statistics for each data set using the default configuration are reported in Table 23.

Table 23: **Detailed statistics about the *default* data sets in `rsbench`**. For generators, the number of concepts $k$ is configurable; in `CLE4EVR`, $n$ and $m$ are the minimum and maximum number of objects.

| TASK | INFO $\mathbf{x}$ | INFO $\mathbf{c}$ | INFO $\mathbf{y}$ | TRAIN | VAL | TEST | OOD |
|---|---|---|---|---|---|---|---|
| MNMath | $28k \times 28$ | $k$ digits, 10 values each | cat multilabel | custom | custom | custom | custom |
| MNAdd-Half | $56 \times 28$ | 2 digits, 10 values each | cat (19 values) | $2,940$ | $840$ | $420$ | $1,080$ |
| MNAdd-EvenOdd | $56 \times 28$ | 2 digits, 10 values each | cat (19 values) | $6,720$ | $1,920$ | $960$ | $5,040$ |
| MNLogic | $28k \times 28$ | $k$ digits, 2 values each | binary | custom | custom | custom | custom |
| Kand-Logic | $3 \times 192 \times 64$ | 3 objects per image
3 shapes
3 colors | binary | $4,000$ | $1,000$ | $1,000$ | – |
| CLE4EVR | $320 \times 240$ | $n$ to $m$ objects per image
10 shapes
10 colors
2 materials
3 sizes | binary | custom | custom | custom | custom |
| BDD-OIA | $1280 \times 720$ | 21 binary concepts | bin multilabel, 4 labels | $16,082$ | $2,270$ | $4,572$ | – |
| SDD-OIA | $469 \times 387$ | 21 binary concepts | bin multilabel, 4 labels | $6,820$ | $1,464$ | $1,464$ | $1,000$ |

## C.1   Licensing

**Code**. Most of our code is available under the `BSD 3-Clause` license. The `CLE4EVR` and `SDD-OIA` generators are derived from the CLEVR code base, which is available under the BSD license. The `Kand-Logic` generator is derived from the `Kandinsky-patterns` code base, which is available under the `GPL-3.0` license, and so is our generator.

**Data**. `MNMath`, `MNAdd-Half`, `MNAdd-EvenOdd` and `MNLogic` are derived from MNIST [63], which is distributed under `CC-BY-SA 3.0`, and so are our data sets and generated data. `BDD-OIA` is derived from `BDD-100k` [108], which is distributed under a `BSD 3-Clause` license, and so is our data set. Data sets and generated data for `Kand-Logic` and `SDD-OIA` are available under a `CC-BY-SA 4.0` license.

## C.2   Ethical Statement

`rsbench` is a collection of datasets aimed at exploring challenges related to concept quality, particularly focusing on identifying reasoning shortcuts. It also includes a formal verification tool to assess how often these shortcuts occur in specific configurations. Essentially, `rsbench` aims to help investigating concept quality in neural, neuro-symbolic and foundation models. Although this is not its intended purpose, such a benchmark may inadvertently used to improve models designed for harmful applications. However, to our knowledge, our work does not directly threaten individuals or society. Additionally, since most datasets are synthetically generated, they do not cause harm during creation. `BDD-OIA`, just like `BDD-100k`, could in principle be used to train models that aim to cause harm. We expressly disapprove of this usage.

## C.3   Hosting and Maintenance Plan

The data is openly available on Zenodo at `https://zenodo.org/doi/10.5281/zenodo.11612555`. The data set generators are freely available on Github. The repository is linked in our website: `https://github.com/unitn-sml/rsbench`.

## C.4 Data Collection

`rsbench` makes uses of two pre-existing data collections, namely `MNIST` and `BDD-OIA`. In this section, we briefly describe this data and how it is collected.

`MNIST`: The `MNIST` [63] dataset is a well known collection of handwritten digits, consisting of $60,000$ training images and $10,000$ test images. Each image is a $28 \times 28$ grayscale image of a numerical digit ranging from $0$ to $9$. The dataset was created by Yann LeCun, Corinna Cortes and Christopher J.C. Burges. `MNAdd-EvenOdd` and `MNAdd-Half` build on the `MNIST` dataset [20, 25]. `MNLogic` and `MNMath`, two datasets that can be generated from `rsbench`, make use of `MNIST` images.

`BDD-OIA`: `BDD-OIA` [19] is a dataset based on `BDD-100K` [108] dataset. `BDD-100K` is a large collection consisting of driving video data, developed by researchers at the University of California, Berkeley. The dataset is suitable for multitask learning, ranging from object detection to semantic segmentation and object tracking. It contains $100,000$ videos and images, collected under diverse driving conditions, times of day, and geographic locations. The data is annotated with labels including bounding boxes, lane marking, and drivable area segmentation. For further information, please refer to the original paper [108].

## C.5 Data Generators

Each `rsbench` data generator comprises two `Python` components: the *generator* proper samples new data, and the associated *parser* reads the configuration from a `YAML` file. The latter also validates the configuration, *i.e.*, check for required fields and ensure the logical formulas work as intended. Users can also configure the generators through the command line. Generated images are stored in `PNG` format, and ground-truth annotations as `JOBLIB` metadata.

**Shared configuration options**. All generators support a set of basic command line settings: `config`: path to the `YAML` configuration file; `output_dir`: path to the output directory; `n_samples`: number of samples to be generated; `log_level`: verbosity level; `seed`: RNG seed, for reproducibility;

They all comply with the following `YAML` settings: `symbols`: names of the logic symbols (concepts) that appear in the knowledge; the order is managed internally by `rsbench`; `logic`: formal specification of the knowledge as a `sympy` formula, used for computing the ground truth labels; `prop_in_distribution`: proportion of examples to put in the in-distribution sets (train, validation, and test), up to $100\%$; `combinations_in_distribution`: what combinations of concept values should be included in the in-distribution sets. `val_prop`: proportion of examples to put in the validation set; `test_prop`: proportion of examples to put in the test set;

**Non-Blender generators**: `MNMath`, `MNLogic`, and `Kand-Logic`. The generator first parses the `YAML` configuration file, then proceeds to randomly sample the required number of examples. It generates a series of label and concept assignments that comply with the combinations combinations specified by the config file, if any. The ground-truth label is computed using the knowledge K. For `MNMath`, which is multi-class and multi-label, this involves splitting the configurations between classes or random sampling. Before the generation of the dataset, `rsbench` automatically checks whether the sampled configurations produce labels that are either all false or all true, and returns an error to the user if such a condition is found.

If the `prop_in_distribution` flag is set, the specified ratio is assigned to the in-distribution datasets (training, validation, and test), while the remaining settings are allocated to the out-of-distribution datasets. An equal number of examples are then assigned to both positive and negative configurations chosen for training, testing, and validation. This is achieved by sampling configurations alternately from positive and negative sides, with replacement. Depending on the dataset, examples are generated, and information such as labels and concepts are stored as `JOBLIB` metadata.

Finally, `rsbench` provides the option to specify a compression type (*e.g.*, zip) for storing the dataset, ensuring efficient storage and easy distribution.

**Blender-based generators**. Generating 3D images involves running scripts from within Blender, which requires a different setup. These scripts read all configuration from the command line and specified configuration files. Options include the positions of shapes (`shape_dir`) and materials (`material_dir`), the output directories (`output_image_dir` for the examples and `output_scene_dir` for metadata), the image resolution (`width`, `height`), and details bout the ren-

dering step (like `render_tile_size`, `render_num_samples`, `camera_jitter`, `light_jitter`). The rendering engine used for CLE4EVR is CYCLES, while SDD-OIA uses the EEVEE rendering engine to speed up rendering, although this can be easily changed by the user.

The generators build on the implementation of [65]. The images are stored as PNGs, while the metadata, in JSON format, contains information about concepts, ground truth labels, object bounding boxes, object positions, and relationships between objects (*e.g.*, that one object is behind another). Unlike the synthetic data generation case, these scripts currently do not offer an option to compress the dataset, though this is a future contribution under consideration.

### C.6  Example usage

rsbench provides functionality for loading, training, and evaluating both the data and models discussed in this paper. This ready-to-use toolkit is available at https://github.com/unitn-sml/rsbench-code/tree/main/rsseval. Alternatively, the data from rsbench can be loaded with minimal code, as demonstrated in the following example:

---
**Listing 1** Code snippet showcasing the training of a neural network on MNLogic using the default configuration.

---
```python
from rss.datasets.xor import MNLOGIC

class required_args:
    def __init__(self):
        self.c_sup = 0 # specifies % supervision available on concepts
        self.which_c = -1 # specifies which concepts to supervise, -1=all
        self.batch_size = 64 # batch size of the loaders

args = required_args()

dataset = MNLOGIC(args)
train_loader, val_loader, test_loader = dataset.get_loaders()

model = #define your model here
optimizer = #define optimizer here
criterion = #define loss function here

for epoch in range(30):
    for images, labels, concepts in train_loader:
        optimizer.zero_grad()
        outputs = model(images)
        loss = criterion(outputs, labels, concepts)
        loss.backward()
        optimizer.step()
```
---

Listing 1 illustrates a typical procedure for training a neural network on MNLogic, following standard PyTorch practices and default configurations.

Additionally, various models and datasets can be employed by providing a script with the appropriate arguments, as shown below:

```
python main.py --dataset mnmathdpl --model mnmathnn --n_epochs 5 --lr 0.001 --seed 8
--batch_size=64 --exp_decay=1 --c_sup 0 --task mnmath
```

Customization of the data and splits is supported, allowing users to explore different experimental settings and corner cases. This customization involves modifying a short JSON or YAML file. Further details and examples can be found in Appendix C.

For formal verification of RSs, rsbench offers a dedicated code base available at https://github.com/unitn-sml/rsbench-code/tree/main/rsscount.

To generate a DIMACS encoding for counting tasks, use the command:

```
python gen-rss-count.py
```

This script supports computing the exact number of RSs for smaller tasks (e.g., XOR with 3 variables) by specifying the `-e` and partial supervision can be specifyied with the `-d` flag. Additionally, random CNFs and custom tasks in DIMACS format are supported. For help with arguments, the `-h` flag is available.

Once the problem encoding is generated, RS counts can be approximated with pyapproxmc using:

```
python count-amc.py PATH --epsilon E --delta D
```

Exact solvers, such as `pyeda` and `pysdd`, can also be employed.

## C.7 `MNMath` Data Generator

Additional `YAML` config for `MNMath` are the number of digits per image (`num_digits`) and the subset of candidate digits (`digit_values`). The code expects `num_digits` names for `symbols`: the first one is assigned to the first digit, the second symbol to the second digit, and so on. With `logic`, the user can provide the system of equations. With `combinations_in_distribution`, the user can have fine-grained control over the in-distribution data (*e.g.*, specifying "0234" means that the in-distribution data contains 𝟬 𝟮 𝟯 𝟰).

**Table 24:** Example of `MNMath` data

| YAML config | JOBLIB metadata | PNG data |
|---|---|---|
| `num_digits: 2`
`symbols:`
`  - a`
`  - b`
`logic:`
`  - 2*a + b`
`  - a + b` | `{`
`    'label': [6, 7],`
`    'meta': {`
`        'concepts': [`
`            [2, 2],`
`            [3, 4]`
`        ]`
`    }`
`}` |  |

## C.8 `MNLogic` Data Generator

The `YAML` file allows to specify the number of Boolean variables in the formula, as well as the formula itself. The knowledge defaults to the $k$-bit `XOR`. `rsbench` includes a script for generating random $\ell$-CNF formulas, which can be readily used with `MNLogic` by setting `xor_rule` to false and `logic` to the target formula. If `use_mnist` is set, the input images are of size $(k \cdot 28) \times 28$ and obtained by concatenating $k$ MNIST digits, one per bit. Otherwise, the code defaults to the setup of [20], where the inputs are encoded as $k \times 1$ black-and-white images, one pixel per bit.

You can filter what types of data appear in-distribution with `combinations_in_distribution` (*e.g.*, specifying 0101 means the in-distribution data contains 𝟬 𝟭 𝟬 𝟭).

## C.9 `Kand-Logic` Data Generator

The `YAML` file allows specifying: `n_shapes`, the number of primitives per figure; `n_figures`: the number of figures per input image; `colors`, a subset of {red, yellow, blue}; `shapes`: a subset of {square, circle, triangle}. The first two `symbols` are associated to the first primitive in the first image, and refer to its shape and color, respectively; the next two to the second primitive, and so on for all primitives and figures in the input. `logic` applies to each individual figure. The ground-truth label of an image (consisting of multiple figures) is specified by `aggregator_symbols` and `aggregator_logic`. These give names to the variables holding the truth value for each figure, and how these values are aggregated to yield the ground-truth label, respectively.

**Table 25:** Example of `MNLogic` data

| YAML config | JOBLIB metadata | PNG data |
|---|---|---|
| `n_digits: 3`
`xor_rule: False`
`symbols:`
`  - a`
`  - b`
`  - c`
`logic:`
`    Or(And(a, b), Not(c))`
`use_mnist: True` | `{`
`  'label': True,`
`  'meta': {`
`    'concepts': [`
`      True,`
`      False,`
`      False`
`    ]`
`  }`
`}` | |

The user can specify which data combination to generate in-distribution by setting `combinations_in_distribution` (*e.g.*, specifying ● "red, square" ● "blue, square" ● "blue, square" means the in-distribution data contains an image made of a red square and two blue squares).

**Table 26:** Example of `Kand-Logic` data

| YAML config | JOBLIB metadata | PNG data |
|---|---|---|
| `colors:`
`  - red`
`  - yellow`
`  - blue`
`shapes:`
`  - circle`
`  - square`
`  - triangle`
`symbols:`
`  - shape_1`
`  - color_1`
`...`
`  - shape_3`
`  - color_3`
`logic:`
`    (Eq(color_1, color_2) &`
`    Eq(shape_1, shape_2) &`
`    Ne(shape_1, shape_3)) |`
`    ... )`
`    # two equal one diff`
`aggregator_symbols:`
`  - pattern_1`
`  - pattern_2`
`  - pattern_3`
`aggregator_logic:`
`    pattern_1 &`
`    pattern_2 &`
`    pattern_3` | `{`
`  'label': True,`
`  'meta': {`
`    'concepts': [`
`      [6, 2,`
`      5, 1,`
`      6, 2],`

`      [6, 1,`
`      5, 2,`
`      6, 1],`

`      [5, 2,`
`      5, 2,`
`      4, 1]`
`    ]`
`  }`
`}` | |

## C.10 `CLE4EVR` Data Generator

The data generation process for `CLE4EVR` closely resembles that of previous datasets. To generate the datasets, the program samples various configurations, specifically the number of objects, shapes,

colors, and sizes. These configurations are then divided into positive and negative sets based on the whether they satisfy the knowledge `logic`. The sets are used to generate images while maintaining a balanced ratio of positive and negative ground-truth samples.

`rsbench` allows users to customize various aspects of data generation, including the number of objects, whether occlusion is permitted, and the dimensions of the image. The occlusion check, which uses Blender rendering, can be slow for many objects due to rejection sampling.

`rsbench` by default includes two materials (rubber and metal), nine shapes, and eight predefined colors, with options to create custom blend files and specify RGB values. Default object sizes are large, medium, and small, but users can fully customize these settings in a configuration file.

The `symbols` for each object, are be defined in the following the order: color, shape, material, and size.

**Table 27:** Example of `CLE4EVR` data

| YAML config | JSON metadata | PNG data |
|---|---|---|
| ```yaml
symbols:
  - color_1
  - shape_1
  - mat_1
  - size_1
  - color_2
  - shape_2
  - mat_2
  - size_2
logic: |
    And(
      Eq(color_1, color_2),
      Eq(shape_1, shape_2),
      Eq(mat_1, mat_2),
      Eq(size_1, size_2)
    )
``` | ```json
{
    "label": 0,
    "concepts": [
    [
      [
        0,
        1,
        0,
        0,
        0,
        0,
        0,
        0
      ],
    ],
}
``` | |

## C.11 `SDD-OIA` Data Generator

$(i)$ Sample label $\qquad$ $(ii)$ Sample concepts $\qquad$ $(iii)$ Sample objects $\qquad$ $(iv)$ Render image

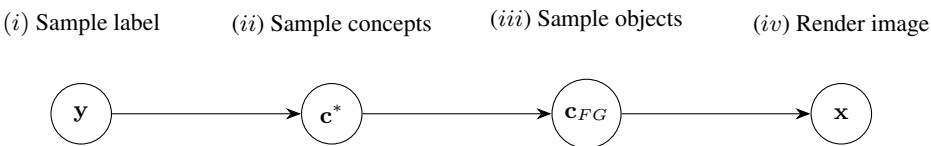

Figure 4: Illustration of the sampling process of `SDD-OIA`

Regarding `SDD-OIA`, `rsbench` allows users to specify parameters such as the number of samples, number of configurations to be generated, and image size.

For `SDD-OIA`, the data generation approach differs from other datasets in `rsbench` and follows a Bayesian network [109]. The process involves first $(i)$ sampling the actions $\mathbf{y}$ from $p(\mathbf{y})$, ensuring that the overall dataset is balanced in the labels, *i.e.*, $p(\mathbf{y})$ is the uniform distribution. $(ii)$ Second, we sample the ground-truth concepts $\mathbf{c}^*$ from the conditional $p(\mathbf{c}^* \mid \mathbf{y})$. Then, $(iii)$ the concepts $\mathbf{c}^*$ specify a fine-grained distribution of objects in the scene, denoted as $\mathbf{c}_{FG}$, which are sampled through $p(\mathbf{c}_{FG}|\mathbf{c}^*)$. Next, the fine-grained objects are used to generate the scene. This step is deterministic and yields the final image $\mathbf{x}$. The crossroads scene is essentially a grid where objects' positions are specified by the fine-grained variables $\mathbf{c}_{FC}$. This ensures the concepts $\mathbf{c}^*$ are visible from the car's camera. The scene is then rendered with blender. The process is shown in Fig. 4. All steps in the sampling procedure ensure that all concepts can be retrieved from the image (respecting

assumption **A1** in Appendix A.3) and that labels can be predicted uniquely from concepts $\mathbf{c}^*$ (respecting assumption **A2** in Appendix A.3).

A key aspect of `SDD-OIA` is its customizable data generation process, which involves sampling the concepts and constructing the scene. This necessitates a hard-coded compositional framework to correctly position the camera and objects, ensuring visibility from the car's perspective. This approach enables the creation of a high-quality synthetic neuro-symbolic dataset, where objects, sample quantities, and distribution ratios are fully customizable. Like other datasets, `SDD-OIA` maintains a balanced distribution across all actions. Users can configure model selection, object dimensions, and the probabilities for sampling different objects by adjusting the categorical distribution weights or the hard-coded matrix configuration.

**Table 28:** Example of `SDD-OIA` data

| JSON metadata | PNG data |
| --- | --- |

```
{
    "label": [
        0,
        1,
        0,
        1
    ],
    "concepts": {
        "red_light": false,
        "green_light": true,
        "car": false,
        "person": false,
        "rider": false,
        "other_obstacle": false,
        "follow": false,
        "stop_sign": false,
        "left_lane": false,
        "left_green_light": true,
        "left_follow": false,
        "no_left_lane": true,
        "left_obstacle": false,
        "left_solid_line": false,
        "right_lane": true,
        "right_green_light": true,
        "right_follow": true,
        "no_right_lane": false,
        "right_obstacle": false,
        "right_solid_line": false,
        "clear": true
    }
}
```

### C.11.1  Assets used in `SDD-OIA`

All assets are made available under permissive licenses that allow reuse for non-commercial purposes.

- Author: stunts. Speed Limit Signs [3D model]. Retrieved from https://free3d.com/3d-model/speed-limit-signs-172903.html;

- Author: corrobrocz. Concrete street barrier [3D model]. Retrieved from https://free3d.com/3d-model/concrete-street-barrier-917223.html;

- Author: paulsendesign. Cartoon low poly trees [3D model]. Retrieved from https://free3d.com/3d-model/cartoon-low-poly-trees-895299.html;

- Author: roxas. Low Poly Car [3D model]. Retrieved from https://free3d.com/3d-model/low-poly-car-14842.html;
- Author: RokoTheAwesome. Traffic Light [3D model]. Retrieved from https://www.turbosquid.com/3d-models/traffic-light-547022

All the models from free3d are under the Personal Use License, meaning the models are available for free but only for personal or non-commercial use. In contrast, the models from TurboSquid are under the Standard 3D Model License, which permits the use of TurboSquid models in various commercial projects, such as games and movies. This license allows the creation and distribution of your end-products without reproduction limitations to any target market or audience indefinitely. However, the license prohibits making the models themselves directly available to end-users, so rsbench redirects to the asset URL.

### C.12 BDD-OIA Data

Data for BDD-OIA are those previously published in [19]. BDD-OIA images are selected from BDD-100k only including franes with complicated scenes where multiple actions {forward, stop, left, right} are possible. This includes situations with multiple objects present. Following [19], all images are manually annotated for ground-truth actions and 21 associated binary concepts. The dataset contains 16k frames for training, (with annotated labels and concepts); 2k frames for validation, and 4.5k frames for testing. The table below reports the overall proportion of labels and concepts.

Concept classes in BDD-OIA

| Action Category | Concepts | Count |
|---|---|---|
| move_forward | green_light | 7805 |
| | follow | 3489 |
| | road_clear | 4838 |
| stop | red_light | 5381 |
| | traffic_sign | 1539 |
| | car | 233 |
| | person | 163 |
| | rider | 5255 |
| | other_obstacle | 455 |
| turn_left | left_lane | 154 |
| | left_green_light | 885 |
| | left_follow | 365 |
| | no_left_lane | 150 |
| | left_obstacle | 666 |
| | letf_solid_line | 316 |
| turn_right | right_lane | 6081 |
| | right_green_light | 4022 |
| | right_follow | 2161 |
| | no_right_lane | 4503 |
| | right_obstacle | 4514 |
| | right_solid_line | 3660 |

# D Additional Results

Here, we report additional tables for `TCAV` evaluation complementing the results reported in the main text. All results indicate that `TCAV` at different layers always attain low $F1$-scores. We also report the $\mathsf{Cls}(C)$ and $\mathrm{mAcc}_C$.

Table 29: Concept metrics for each `NN` layer using TCAV on `MNAdd-EvenOdd`

|  | LAYER NUM | $\mathrm{Acc}_C$ | $F_1(C)$ | $\mathsf{Cls}(C)$ |
|---|---|---|---|---|
| $conv_1$ | 1 | $0.11 \pm 0.03$ | $0.10 \pm 0.03$ | $0.00 \pm 0.00$ |
| $conv_2$ | 2 | $0.12 \pm 0.03$ | $0.10 \pm 0.04$ | $0.01 \pm 0.02$ |
| $fc_1$ | 3 | $0.12 \pm 0.04$ | $0.09 \pm 0.05$ | $0.24 \pm 0.30$ |
| $fc_2$ | 4 | $0.11 \pm 0.02$ | $0.07 \pm 0.03$ | $0.29 \pm 0.34$ |

Table 30: Concept metrics for each `NN` layer using TCAV on `Kand-Logic`

|  | LAYER NUM | $\mathrm{Acc}_C$ | $F_1(C)$ | $\mathsf{Cls}(C)$ |
|---|---|---|---|---|
| $conv1$ | 1 | $0.35 \pm 0.01$ | $0.34 \pm 0.01$ | $0.00 \pm 0.01$ |
| $conv2$ | 2 | $0.35 \pm 0.01$ | $0.34 \pm 0.01$ | $0.00 \pm 0.01$ |
| $conv3$ | 3 | $0.34 \pm 0.01$ | $0.34 \pm 0.01$ | $0.00 \pm 0.01$ |
| $conv4$ | 4 | $0.35 \pm 0.01$ | $0.34 \pm 0.01$ | $0.00 \pm 0.01$ |
| $conv5$ | 5 | $0.35 \pm 0.01$ | $0.34 \pm 0.01$ | $0.00 \pm 0.01$ |
| $fc1$ | 6 | $0.33 \pm 0.01$ | $0.32 \pm 0.01$ | $0.00 \pm 0.01$ |
| $fc2$ | 7 | $0.33 \pm 0.01$ | $0.31 \pm 0.01$ | $0.00 \pm 0.01$ |

Table 31: Concept metrics for each `NN` layer using TCAV on `SDD-OIA`

|  | LAYER NUM | $\mathrm{mAcc}_C$ | $\mathrm{mF}_1(C)$ | $\mathsf{Cls}(C)$ |
|---|---|---|---|---|
| $conv1$ | 1 | $0.48 \pm 0.02$ | $0.44 \pm 0.01$ | $0.19 \pm 0.05$ |
| $conv2$ | 2 | $0.49 \pm 0.02$ | $0.45 \pm 0.02$ | $0.20 \pm 0.06$ |
| $conv3$ | 3 | $0.49 \pm 0.03$ | $0.45 \pm 0.03$ | $0.21 \pm 0.09$ |
| $conv4$ | 4 | $0.48 \pm 0.02$ | $0.44 \pm 0.01$ | $0.23 \pm 0.15$ |
| $conv5$ | 5 | $0.48 \pm 0.02$ | $0.44 \pm 0.02$ | $0.30 \pm 0.26$ |
| $conv6$ | 6 | $0.46 \pm 0.02$ | $0.43 \pm 0.02$ | $0.34 \pm 0.33$ |
| $fc1$ | 7 | $0.50 \pm 0.02$ | $0.45 \pm 0.03$ | $0.38 \pm 0.31$ |
| $fc2$ | 8 | $0.49 \pm 0.02$ | $0.44 \pm 0.02$ | $0.43 \pm 0.28$ |

Table 32: Concept metrics for each `NN` layer using TCAV on `SDD-OIA` with synthetic images.

|  | LAYER NUM | $\mathrm{mAcc}_C$ | $\mathrm{mF}_1(C)$ | $\mathsf{Cls}(C)$ |
|---|---|---|---|---|
| $conv1$ | 1 | $0.47 \pm 0.02$ | $0.43 \pm 0.02$ | $0.18 \pm 0.03$ |
| $conv2$ | 2 | $0.48 \pm 0.02$ | $0.44 \pm 0.02$ | $0.18 \pm 0.03$ |
| $conv3$ | 3 | $0.49 \pm 0.01$ | $0.45 \pm 0.01$ | $0.23 \pm 0.12$ |
| $conv4$ | 4 | $0.48 \pm 0.03$ | $0.44 \pm 0.03$ | $0.23 \pm 0.14$ |
| $conv5$ | 5 | $0.48 \pm 0.02$ | $0.44 \pm 0.02$ | $0.29 \pm 0.25$ |
| $conv6$ | 6 | $0.48 \pm 0.04$ | $0.45 \pm 0.04$ | $0.34 \pm 0.32$ |
| $fc1$ | 7 | $0.51 \pm 0.03$ | $0.45 \pm 0.03$ | $0.38 \pm 0.31$ |
| $fc2$ | 8 | $0.74 \pm 0.01$ | $0.42 \pm 0.01$ | $0.99 \pm 0.01$ |

# E  Dataset Documentation: Datasheets for Datasets

Here, we answer the questions posed in the datasheets for datasets paper by Gebru et al [110].

## E.1  Motivation

**For what purpose was the dataset created?**  `rsbench` was created to study the phenomenon of reasoning shortcuts (RSs) and concept quality in neuro-symbolic and neural architectures. `rsbench` offers several datasets where RSs occur, as well as a formal verification tool that enables users to verify how many RSs appear in the desired settings.

**Who created the dataset (*e.g.*, which team, research group) and on behalf of which entity (*e.g.*, company, institution, organisation)?**  The datasets have been created by the "Structured Machine Learning" research group at the department of Information Engineering and Computer Science of the University of Trento in collaboration with the april Lab at School of Informatics, University of Edinburgh.

**Who funded the creation of the dataset?**  The datasets have been created for research purposes. Funded by the European Union. The views and opinions expressed are however those of the author(s) only and do not necessarily reflect those of the European Union, the European Health and Digital Executive Agency (HaDEA) or the European Research Executive Agency. Neither the European Union nor the granting authority can be held responsible for them. Grant Agreement no. 101120763 - TANGO. PM is supported by the MSCA project GA nř101110960 Probabilistic Formal Verification for Provably Trustworthy AI - PFV-4-PTAI. AV is supported by the "UNREAL: Unified Reasoning Layer for Trustworthy ML" project (EP/Y023838/1) selected by the ERC and funded by UKRI EPSRC. Emile van Krieken was funded by ELIAI (The Edinburgh Laboratory for Integrated Artificial Intelligence), EPSRC (grant no. EP/W002876/1).

## E.2  Composition

**What do the instances that comprise the dataset represent (*e.g.*, documents, photos, people, countries)?**  All datasets contain annotations regarding concepts and labels. `SDD-OIA` comprises synthetically generated images depicting autonomous driving scenarios, such that if they were captured from a car's dashcam, and includes additional information about the scene structure, such as bounding boxes, 2D and 3D coordinates, and spatial relationships among objects. `MNMath`, `MNAdd-Half`, `MNAdd-EvenOdd` and `MNLogic` contain synthetic images of handwritten digits, derived from the `MNIST` dataset. `Kand-Logic` consists of synthetic data showcasing patterns of geometric shapes with various colors. `CLE4EVR` features synthetically generated images representing 3D objects of different shapes, colors, materials, and dimensions; similar to `SDD-OIA`, they include additional scene information. `BDD-OIA` is a real-world, high-stakes dataset comprising images captured from a car's dashcam. For a comprehensive description, please refer to [19].

**How many instances are there in total (of each type, if appropriate)?**  Please refer to Table 23.

**Does the dataset contain all possible instances or is it a sample (not necessarily random) of instances from a larger set?**  The datasets represent samples from configurations that can be randomly generated according to a grammar. Using the generators, one can filter through various combinations and determine the level of exhaustiveness for generating examples. For a comprehensive overview of each dataset generation process, please consult Appendix C.5 and subsequent sections.

**What data does each instance consist of?**  Alongside the images, each dataset sample is annotated with concepts and labels. However, for `SDD-OIA` and `CLE4EVR`, detailed scene information is included, encompassing individual 2D and 3D coordinates, bounding boxes, and spatial relationships between objects. For an complete overview refer to Table 23.

**Is there a label or target associated with each instance?**  Yes, the concept annotations are derived from the data generation process, while the labels are symbolically derived from the knowledge provided to the dataset.

**Is any information missing from individual instances?**    No.

**Are relationships between individual instances made explicit (*e.g.*, users' movie ratings, social network links)?**    No, there are no connections between different instances.

**Are there recommended data splits (*e.g.*, training, development/validation, testing)?**    Information about the data splits we employed is reported in Appendix B. The user has the freedom to choose the data splits they prefer during the data generation process.

**Are there any errors, sources of noise, or redundancies in the dataset?**    No.

**Is the dataset self-contained, or does it link to or otherwise rely on external resources (*e.g.*, websites, tweets, other datasets)?**    Some of our data sets build on top of established and stable data, namely MNIST and (the last frames provided by) BDD-100k, for which we provide download links. SDD-OIA makes use of external assets, listed in Appendix C.11.1. The ready-made SDD-OIA data set does not require these assets, but in order to use the generator these have to be obtained separately.

**Does the dataset contain data that might be considered confidential (e.g., data that is protected by legal privilege or by doctor-patient confidentiality, data that includes the content of individuals' non-public communications)?**    No.

**Does the dataset contain data that, if viewed directly, might be offensive, insulting, threatening, or might otherwise cause anxiety?**    No.

**Does the dataset relate to people? If not, you may skip the remaining questions in this section.**
BDD-OIA contains images depicting pedestrians and bicycle riders. Identifiable information in these images, including anonymization, rights, and risks, is managed by the original BDD-100k authors.

**Does the dataset identify any subpopulations (*e.g.*, by age, gender)?**    Please refer to E.2.

**Is it possible to identify individuals (*i.e.*, one or more natural persons), either directly or indirectly (*i.e.*, in combination with other data) from the dataset?**    Please refer to E.2.

**Does the dataset contain data that might be considered sensitive in any way (e.g., data that reveals racial or ethnic origins, sexual orientations, religious beliefs, political opinions or union memberships, or locations; financial or health data; biometric or genetic data; forms of government identification, such as social security numbers; criminal history)?**    Please refer to E.2.

### E.3   Collection Process

**How was the data associated with each instance acquired?**    MNIST and BDD-100k have been obtained from their official repositories, http://yann.lecun.com/exdb/mnist/ and https://dl.cv.ethz.ch/bdd100k/data/, respectively. All other data is synthetically generated.

**What mechanisms or procedures were used to collect the data (*e.g.*, hardware apparatus or sensor, manual human curation, software program, software API)?**    Details about data generations and software programs are discussed in Appendix B.

**If the dataset is a sample from a larger set, what was the sampling strategy (*e.g.*, deterministic, probabilistic with specific sampling probabilities)?**    Please refer to the similar question in Appendix E.2.

**Who was involved in the data collection process (*e.g.*, students, crowdworkers, contractors) and how were they compensated (*e.g.*, how much were crowdworkers paid)?**    The authors were involved in the process of generating these datasets.

**Over what timeframe was the data collected?**   The datasets were generated over a span of several days.

**Were any ethical review processes conducted (*e.g.*, by an institutional review board)?**   No.

**Does the dataset relate to people? If not, you may skip the remainder of the questions in this section.**   BDD-OIA is the only dataset relating to people, please refer to Appendix E.2.

## E.4  Preprocessing/Cleaning/Labeling

**Was any preprocessing/cleaning/labeling of the data done (*e.g.*, discretization or bucketing, tokenization, part-of-speech tagging, SIFT feature extraction, removal of instances, processing of missing values)?**   No, the datasets were generated along with labels and concept annotations.

**Was the "raw" data saved in addition to the preprocessed/cleaned/labeled data (*e.g.*, to support unanticipated future uses)?**   NA

**Is the software used to preprocess/clean/label the instances available?**   NA

## E.5  Uses

**Has the dataset been used for any tasks already?**   In the paper, we demonstrate and benchmark the intended use of these datasets for evaluating concept quality and exploring RSs. MNAdd-EvenOdd, MNAdd-Half, and CLE4EVR have been utilized in previous studies [25, 20, 17] to investigate RSs and concept quality.

**Is there a repository that links to any or all papers or systems that use the dataset?**   Yes, https://unitn-sml.github.io/rsbench/.

**What (other) tasks could the dataset be used for?**   SDD-OIA and CLE4EVR offer additional information regarding the scene, including the 3D and 2D coordinates of objects, their bounding boxes, and the relationships between objects within the scene. This spatial data enables various applications such as object discovery, object detection, and reasoning over the scene's structure.

**Is there anything about the composition of the dataset or the way it was collected and preprocessed/cleaned/labeled that might impact future uses**   No.

**Are there tasks for which the dataset should not be used?**   These datasets are meant for research purposes only.

## E.6  Distribution

**Will the dataset be distributed to third parties outside of the entity (*e.g.*, company, institution, organization) on behalf of which the dataset was created?**   No.

**How will the dataset will be distributed (*e.g.*, tarball on website, API, GitHub)?**   The datasets, data generators, and related evaluation code are available on the website, enabling users to generate, download, and test their model on the data. Each dataset is provided in zip format and can be downloaded from the Zenodo link on the website.

**When will the dataset be distributed?**   The datasets employed in the paper are available now on the website.

**Will the dataset be distributed under a copyright or other intellectual property (IP) license, and/or under applicable terms of use (ToU)?**   Please refer to Appendix C.1.

**Have any third parties imposed IP-based or other restrictions on the data associated with the instances?** SDD-OIA makes use of assets taken from https://free3d.com and https://www.turbosquid.com. See Appendix C.11.1 for the full list and associated licenses. Other instances of datasets themselves do not have IP-based restrictions.

**Do any export controls or other regulatory restrictions apply to the dataset or to individual instances?** Not that we are are of.

## E.7 Maintenance

**Who is supporting/hosting/maintaining the dataset?** The datasets are supported by the authors and will be actively maintained by the "Structured Machine Learning" research group in the future. For the hosting and maintenance plan, please refer to Appendix C.3.

**How can the owner/curator/manager of the dataset be contacted (e.g., email address)?** The authors of rsbench can be contacted via their email addresses: samuele.bortolotti@unitn.it, emanuele.marconato@unitn.it.

**Is there an erratum?** If errors are found, an erratum will be added to the website.

**Will the dataset be updated (*e.g.*, to correct labeling errors, add new instances, delete instances)?** Any potential future updates or extensions will be communicated via the website. The datasets will be versioned.

**If the dataset relates to people, are there applicable limits on the retention of the data associated with the instances (*e.g.*, were individuals in question told that their data would be retained for a fixed period of time and then deleted)?** The only dataset involving people is BDD-OIA, plase refer to Appendix E.2.

**Will older versions of the dataset continue to be supported/hosted/maintained?** We plan to continue hosting older versions of the dataset.

**If others want to extend/augment/build on/contribute to the dataset, is there a mechanism for them to do so?** Yes, the dataset generation code is available on our website.

## E.8 Other Questions

**Is your dataset free of biases?** Our data sets are designed to induce a particular type of bias, namely reasoning shortcuts, in models, for the purpose of studying them. The data itself however is not biased towards human factors such as gender, ethnicity, age, etc.

**Can you guarantee compliance to GDPR?** No, we are unable to comment on legal matters.

## E.9 Author Statement of Responsibility

The authors assume full responsibility for any rights violations and confirm the license associated with the datasets and their images.

