# OpenReview forum: "A Neuro-Symbolic Benchmark Suite for Concept Quality and Reasoning Shortcuts"
_NeurIPS.cc/2024/Datasets_and_Benchmarks_Track — NeurIPS 2024 Track Datasets and Benchmarks Poster_

### Official Review · Reviewer_RJC8 · 2024-07-12
**A collection of tasks for benchmarking reasoning shortcuts**

**Rating:** 7
**Confidence:** 3
**Correctness:** Experiment design and methods seem co…
**Clarity:** Paper is well written.

**Review:**

The paper is easy to follow and provides enough background for people not familiar with reasoning shortcuts to understand the content and contributions. I learned something new and enjoyed reading the paper. My main critic is that the benchmark was not applied to all tasks but only three (MNAdd-EvenOdd, Kand-Logic, SDD.OIA). Even more, only one of these tasks is new (SDD-OIA). It is unclear why the authors present new tasks but do not evaluate their model on them.

**Strengths:**

The paper is easy to follow and understandable without prior knowledge about RS. It is also straight forward to see what are the authors contributions and what was taken from literature.

**Additional Feedback:**

I think there is a missing "to" in the sentence "suggests that NeSy models tend mix concepts together"

**Documentation:**

Code is provided on GitHub and except for the TCAV part everything was clear to me.

**Ethics:**

No concerns.

**Limitations:**

My main critic is that the benchmark was not applied to all eight tasks but only three (MNAdd-EvenOdd, Kand-Logic, SDD.OIA).

**Opportunities For Improvement:**

My main concerns is that the authors introduce three new benchmark tasks but do not evaluate models on two of them. This would be mandatory for the paper to be accepted.

Apart from that, I am having difficulties to see how TCAV is utilized to visualize concepts for black-box models. It is briefly mentioned in the appendix, but it did not really help me. It could be interesting to visualize the concepts extracted by TCAV.

**Relation To Prior Work:**

The authors rely on prior tasks but clarify the changes they made.

**Summary And Contributions:**

The paper presents rsbench, a collection of existing and new tasks for benchmarking reasoning shortcuts.  In total, 8 tasks from arithmetic, logic and high-stake settings are provided.
The tasks are listed below. For new tasks developed by the authors a description is included:

- MNMath (new) is a "multi-label extension of MNAdd in which the goal is to predict the result of a system of equations of MNIST digits"
- MNAdd-Half
- MNAdd-EvenOdd

- MNLogic (new) is "a logical analogue to MNAdd in which inference is driven by a random logic formula"
- Kand-Logic
- CLE4EVR

- BDD-OIA
- SDD-OIA (new) is " a synthetic replacement for BDD-OIA that comes with a fully configurable data generator, enabling fine-grained control over what labels, concepts, and images are observed and the creation of OOD split"


For each task, either an existing dataset is provided or a generator to create the data.
The benchmark further contains three existing metrics for evaluation and a novel algorithm for counting shortcuts referred to as countrss.

The three tasks of the benchmark (MNAdd-EvenOdd, Kand-Logic, SDD-OIA) are applied to five models (DPL, LTN, CBM, NN, CLIP).

---

> ### Author Rebuttal · Authors · 2024-08-17
>
> We thank the reviewer for their positive feedback, we are glad they found our paper easy to follow and appreciated our contribution. Below, we briefly address their remarks:
>
> > Evaluating the two new Nesy tasks
> Our experiments were meant to evaluate the ability of the rsbench tasks to induce and assess RSs in different models.
>
> We follow the reviewer's concern and evaluate RSs in the two new tasks: MNMath and MNLogic.
> In short: as expected, all tested models (DPL, NN, CBM) suffer from poor concept quality (low F1 score and medium to high collapse) while attaining high scores on label predictions.  We plan to report the full results in the paper and the supplementary.  See below for details.
>
> - **MNMath**: the input consists of $8$ MNIST digits concatenated into a single image  $\mathbf x = (\mathbf x_1, \mathbf x_2, \mathbf x_3, \mathbf x_4)$. The goal is to predict the result of a system of two equations.  The default system, which we evaluated, consists of two equations: $\mathbf x_1 + \mathbf x_2 = \mathbf x_3 + \mathbf x_4$ and $\mathbf x_5 \times \mathbf x_6 = \mathbf x_7 \times \mathbf x_8$. The results in Table 1 below indicate that, as expected, this task induces RSs in DPL, NN, and CBM: label accuracy is acceptable, while concept quality is very low.
>
> |          	| Acc Y (^)           	| F1 Y (^)          	| Acc C (^)           	| F1 C (^)          	| Collapse C (v)            	|
> |--------------|-------------------------|-----------------------|-------------------------|-----------------------|-------------------------------|
> | DPL      	|  0.80  +/- 0.10     	|  0.73  +/- 0.13   	|  0.11  +/- 0.01     	|  0.03  +/- 0.02   	|  0.01  +/- 0.01           	|
> | CBM      	|  0.75  +/- 0.01     	|  0.67  +/- 0.01   	|  0.22  +/- 0.04     	|  0.11  +/- 0.03   	|  0.68  +/- 0.15           	|
> | NN       	|  0.75  +/- 0.01     	|  0.67  +/- 0.01   	|  0.10  +/- 0.01     	|  0.03  +/- 0.01   	|  0.80  +/- 0.11           	|
>
> **Table 1**: Results on MNISTMATH
>
> - **MNISTLogic**: the default formula is a XOR operation on $4$ MNIST digits restricted to $0$ and $1$. Like abive, the raw input is the concatenation of four MNIST digits $\mathbf x = (\mathbf x_1, \mathbf x_2, \mathbf x_3, \mathbf x_4)$ and the knowledge specifies that the label is the result of $ \mathbf x_1 \oplus \mathbf x_2 \oplus \mathbf x_3 \oplus \mathbf x_4)$. We use $4$ separate neural networks to learn the bits separately. Our results are shown below for DPL, NN, and CBM:
>
> |          	| Acc Y (^)           	| F1 Y (^)          	| Acc C (^)           	| F1 C (^)          	| Collapse C (v)            	|
> |--------------|-------------------------|-----------------------|-------------------------|-----------------------|-------------------------------|
> | DPL      	|  0.99  +/- 0.01     	|  0.99  +/- 0.01   	|  0.51  +/- 0.06     	|  0.47  +/- 0.05   	|  0.01  +/- 0.01           	|
> | CBM      	|  0.95  +/- 0.10     	|  0.89  +/- 0.23   	|  0.50  +/- 0.05     	|  0.48  +/- 0.05   	|  0.01  +/- 0.01           	|
> | NN       	|  0.98  +/- 0.01     	|  0.60  +/- 0.20   	|  0.46  +/- 0.05     	|  0.41  +/- 0.10   	|  0.10  +/- 0.20           	|
>
> **Table 2**: Results on MNISTLogic
>
> > How is TCAV used?
>
> We report additional details on TCAV which we plan to add in a dedicated section in the appendix. We follow the original implementation of TCAV [48 in the paper]:
>
> 1. Concept Activation Vectors (CAVs): For each concept, we train a linear separator in the model’s embeddings space to discriminate between inputs in which a concept is present $x_c$ and inputs in which it is not $x_{\neg c}$. Afterwards, we extract the weights of the linear decision boundary $v_{\text{CAV}}$.
>
> 2. TCAV Score:  The TCAV score is used to measure the presence of a concept. Specifically, we check whether the model's prediction aligns positively with the CAV:
>
>    $$\frac{\partial f(x_i)}{\partial h(x_i)} \cdot v_{\text{CAV}}$$
>
> Here, $\frac{\partial f(x_i)}{\partial h(x_i)}$ is the gradient of the model's output in embedding space $h(x_i)$ and $\cdot$ denotes the dot product.
>
> More in detail:
>
> - For all MNIST-based tasks, CAVs are computed for digits 0-9, yielding $n \times 10$ distinct datasets (in a one vs all fashion, i.e., digit is present vs. digit is not present). Predictions are obtained by selecting, for each position, the digit that has the highest (positive) score.
>
> - Kandinsky Patterns: CAVs are used to identify different colors and shapes. For each figure, the TCAV score is split into color and shape, with predictions based on the highest (positive) scoring CAVs.
>
> - BDD-OIA and SDD-OIA: As the concepts in these datasets are multilabel, we determine the presence of a concept if its TCAV score is positive.
>
> We will make sure to clarify the procedure in the supplementary material (Appendix A.2).
>
> We are keen to clarify any remaining doubts.

---

> > ### Author Response · Authors · 2024-08-26
> >
> > Dear Reviewer,
> >
> > we would be grateful if you could let us know whether we have managed to answer
> > all your questions, and whether there are any remaining doubts.
> >
> > Thank you!
> > The Authors

---

> > > ### Comment · Reviewer_RJC8 · 2024-08-26
> > >
> > > Thank you for taking the time and providing the explanation and new experiments. Since my main concern was regarding the two missing benchmarks, I have increased my rating.

---

### Official Review · Reviewer_tb9x · 2024-07-24

**Rating:** 7
**Confidence:** 3
**Correctness:** All the claims made in the submission…
**Clarity:** The paper explains the proposed task …

**Review:**

The proposed **rsbench** provides customizable tasks affected by RSs, metrics for evaluating concept quality, and novel verification procedures to assess the presence of RSs. It includes various tasks requiring both arithmetical and logical reasoning, and high-stakes tasks such as autonomous driving scenarios. The suite supports purely neural, neuro-symbolic, and concept-bottleneck models, facilitating the evaluation of different architectures and mitigation strategies.

**Strengths:**

1. **rsbench** offers a wide range of tasks, including arithmetic, logic, and high-stakes scenarios, providing a thorough evaluation of models across different domains.
2. The benchmark allows users to create custom tasks and datasets, enabling fine-grained control over training, validation, and test splits.
3. Introduces new formal verification methods to identify and count RSs, improving the assessment of model reliability.
4. Evaluates purely neural, neuro-symbolic, and concept-bottleneck models, making it versatile for various research approaches.

**Additional Feedback:**

NA

**Documentation:**

Yes. The settings and configuration are very complex to use, making it hard to be usable for future research.

**Limitations:**

The authors have adequately addressed the limitations and potential negative societal impact of their work

**Opportunities For Improvement:**

1. The comprehensive nature and customizability might overwhelm new users, requiring a steep learning curve to fully utilize the benchmark.
2. While it includes a few high-stakes tasks, the range might be insufficient for some real-world applications, necessitating further expansion.
3. Some tasks require a modicum of concept-level supervision, which might not always be available or practical in real-world settings.
4. The need to generate and manage multiple custom datasets and configurations might introduce significant overhead in terms of computational resources and time.

**Relation To Prior Work:**

The authors reviewed related works in the last 5 years.

**Summary And Contributions:**

Reasoning shortcuts occur when models solve reasoning tasks without associating the correct concepts to high-dimensional data, undermining the purpose of such models, especially in high-stakes scenarios.
The paper introduces **rsbench**, a benchmark suite designed to systematically evaluate reasoning shortcuts (RSs) in models that combine learning and reasoning.

---

> ### Author Response · Authors · 2024-08-17
> **Dummy**
>
> Empty

---

> ### Author Rebuttal · Authors · 2024-08-17
>
> We thank the reviewer for finding our paper clear and for the versatility of our suite. We reply to points raised by the reviewer.
>
> > Usability of our suite
>
> We agree that, at a first sight, our suite may seem complicated to use. However, **all rsbench tasks come with default settings** (comprising concrete datasets, splits, and data loaders) that can be readily used for training and evaluation. We envision this will be the main usage of rsbench for regular users.
>
> For reference, here is a code snippet showcasing training a neural network on MNLogic using the default configuration:
>
> ```python
>
> from rss.datasets.xor import MNLOGIC
>
> class required_args:
> 	def __init__(self):
> 	self.c_sup = 0 # specifies % supervision available on concepts
> self.which_c = -1 # specifies which concepts to supervise, -1=all
> self.batch_size = 64 # batch size of the loaders
>
> args = required_args()
>
> dataset = MNLOGIC(args)
> train_loader, val_loader, test_loader = dataset.get_loaders()
>
> model = #define your model here
> optimizer = #define optimizer here
> criterion = #define loss function here
>
> for epoch in range(30):
> 	for images, labels, concepts in train_loader:
>     	optimizer.zero_grad()
>     	outputs = model(images)
>     	loss = criterion(outputs, labels, concepts)
>     	loss.backward()
>     	optimizer.step()
> ```
>
> As can be seen, this is rather standard pytorch code.
>
> Additionally, we provide inside the `rss` folder  a ready-on implementation of different models and datasets that can be used by simply providing a script with relevant arguments (see also the documentation):
>
> ```sh
> python main.py --dataset mnmathdpl --model mnmathnn --n_epochs 5 --lr 0.0001 --seed 8 --batch_size=64 --exp_decay=1 --c_sup 0 --task mnmath
> ```
>
> This script – which also consists of standard pytorch code – also serves as a reference for more advanced users.
>
> Customization (of the data and splits) is a way to **enable advanced users to test specific settings and corner cases**. We view this as a plus. Moreover, customization amounts to updating a short JSON/YAML file and requires no modifications to the data loaders, details and examples are provided in Appendix C.
>
> We will to clarify these points in Section 3; we will also include a **how to guide** in the paper and on the website covering how to 1) Configure and run the data generators, 2) Load the data, 3) Train a model, 4) Evaluate a model.
>
> Please let us know if you have any remaining concerns.
>
> > Concept-level supervision
>
> In NeSy, models are not trained with concept supervision, and rsbench works just fine when no concept supervision is provided.
>
> In our evaluation, we only provide concept supervision to CBMs, and only because in absence of concept supervision CBMs become indistinguishable from regular neural networks.
>
> We will clarify this point in the Experiments section.
>
> > Extension to other high-stakes dataset
>
> Besides BDD-OIA and SDD-OIA, as mentioned in Section 5, we plan to integrate ROAD-R in the future. ROAD-R is a high-stakes task that consists of autonomous driving videos annotated with bounding boxes of objects and persons in the scene. It also comprises logical relationships between objects, which makes it suitable for NeSy. However, the complexity of the video processing and the extensive computational resources required put it beyond our reach for the rebuttal.
>
> We point that the lack of available high-stakes datasets is a concern in NeSy at large, where much of the focus has been on semi-synthetic datasets, and also in CBMs, where typical datasets are CUB200 and CelebA.
>
> We will definitely integrate additional high-stakes dataset in rsbench as they become available.

---

> > ### Author Response · Authors · 2024-08-26
> >
> > Dear Reviewer,
> >
> > we would be grateful if you could let us know whether we have managed to answer
> > all your questions, and whether there are any remaining doubts.
> >
> > Thank you!
> > The Authors

---

> > > ### Comment · Reviewer_tb9x · 2024-08-27
> > > **reply**
> > >
> > > Thanks for addressing my concern. I'm happy to improve my score.

---

### Official Review · Reviewer_NquB · 2024-07-24

**Rating:** 7
**Confidence:** 4
**Correctness:** The claims made by the paper appear t…
**Clarity:** The paper is overall well-written, wi…

**Review:**

The paper presents a comprehensive suite of benchmarks evaluating the impact of spurious correlations on models across a variety of tasks, from arithmetic and logic-based tasks to advanced practical tasks like autonomous driving. Notably, the self-driving tasks employ real-world autonomous driving datasets (BDD-OIA) as well as a synthetic framework (SDD-OIA) to generate any imaginable scenarios for RS assessment. Systematic evaluation metrics are introduced, ranging from model-level metrics like concept-level confusion matrices and concept collapse to task-level metrics such as the countrss algorithm. Using these datasets, benchmarks, tasks, and algorithms, the paper extensively evaluates common NeSy models and traditional neural networks in terms of their out-of-distribution (OOD) performances. Incorporating additional reasoning task variants, such as Sudoku and Raven's Progressive Matrices (RPMs), would enhance the comprehensiveness of the benchmarks. To address the complexity of interpreting extensive RS evaluation results, developing interactive dashboards and visualization tools that provide real-time feedback on the impact of RSes on models would be beneficial. These tools would help researchers intuitively understand and refine models, making the paper's contributions even more significant. Overall, the work is of high quality, clear, original, and holds substantial potential for advancing the evaluation of OOD robustness in machine learning models.

**Strengths:**

1. A comprehensive suite of benchmarks focusing on evaluating the impact of RSes on models with a variety of tasks, ranging from arithmetic-based, logic-based, to more advanced and practical tasks such as autonomous driving.
1. Aside from employing real-world autonomous driving datasets, BDD-OIA, to form one of the RS-assessing tasks, the author went a step further and created a synthetic version of the self-driving dataset, SDD-OIA, such that any scenarios imaginable can be generated to satisfy any demands required when assessing RSes.
1. The paper introduces systematic evaluation metrics to gauge the levels of impact of RSes on target models, ranging from model-level metrics that employ concept-level confusion matrices and concept collapse to gauge the extent, to task-level metrics, i.e., the countrss algorithm that counts the RSes exhibited by target knowledge rules/models.
1. Using the established datasets, benchmarks, tasks, and algorithms, the paper extensively evaluates common NeSy models as well as plain-old neural networks in terms of their OOD performances.

**Additional Feedback:**

I have no additional comments.

**Documentation:**

Documentations regarding the dataset and codebase are adequately provided in the main paper, the website linked in the paper, and the released codebase READMEs.

**Ethics:**

There appear to be no notable ethical concerns regarding this work due to the nature of the proposed datasets (patterns and synthetic).

**Limitations:**

The authors have addressed the limitations of their work in Section 5 and in the appendices.

**Opportunities For Improvement:**

As highlighted in the paper, incorporating additional reasoning task variants such as Sudoku and Raven's Progressive Matrices (RPMs) would significantly enhance the comprehensiveness of the benchmark suite. Furthermore, interpreting the extensive reasoning shortcut (RS) evaluation results and systematically visualizing their impact on models could be challenging and overwhelming for future researchers. To address this, it would be beneficial to develop interactive dashboards and visualization tools. These tools should provide real-time feedback on the impact of RSes on models, including concept quality, confusion matrices, and countrss results. Such enhancements would facilitate a more intuitive and efficient understanding of how RSes affect various aspects of model performance, ultimately aiding researchers in refining their models more effectively.

**Relation To Prior Work:**

The paper clearly discussed its contributions in relation to the prior work Sections 1 and 5.

**Summary And Contributions:**

This work aims to tackle the issue of reasoning shortcuts (RS) that most, if not all, common neural classifiers suffer from. RS, if left unhandled or unbounded, may cause reasoning modules to associate unintended features with their decision-making process, thus hindering its out-of-distribution (OOD) abilities, robustness, safety, etc. To mitigate the problem of RS, the authors of this work introduce the rsbench benchmark suite that aims to provide systematic evaluations of the impact of RS on model performances by providing a collection of customizable tasks that are provably affected by RSes with datasets and generators for various OOD scenarios, and a set of quality metrics for organized assessments of the RSes’ impact on the NeSy models. Furthermore, the paper introduces a novel algorithm, countrss, that verifiably determines whether a task performed by a particular model was indeed affected by RSes and counts them.

---

> ### Author Response · Authors · 2024-08-17
> **Dummy**
>
> Empty

---

> ### Author Rebuttal · Authors · 2024-08-17
>
> We thank the reviewer for the positive comments, in particular for finding our paper high-quality, well-written, important, and for appreciating the introduction of SDD-OIA.  In the following, we address their remarks:
>
> > Extension to new datasets
>
> We fully agree with the reviewer. We aim to integrate new datasets in the near future. Unfortunately, including them in time for the rebuttal is not straightforward:
>
> 1. Encoding a sudoku board requires 81 categorical concepts that interact via long-range constraints, and as such many SOTA NeSy architectures struggle with Sudoku [1]; we need to investigate what architectures are best suited for it.
>
> 2. Raven Progressive Matrices has only been considered in a custom NeSy model [2] and it is not clear how to adapt it to architectures like DeepProbLog and Logic Tensor Networks. We hope that the introduction of unified NeSy frameworks like ULLER [3] will facilitate adding such custom architectures to our implementation.
>
> [1] Cristina Cornelio et al., Learning where and when to reason in neuro-symbolic inference, ICLR (2023).
>
> [2] Hersche et al. A Neuro-vector-symbolic Architecture for Solving Raven's Progressive Matrices, Nature Machine Intelligence (2023).
>
> [3] Van Krieken et al., ULLER: A Unified Language for Learning and Reasoning, arXiv (2024)
>
> > Visualization tools
>
> Great point, having a unified dashboard / visualization tool is definitely part of our plan.
>
> Unfortunately, it is not something that we can develop for the rebuttal, primarily because there is no shared library for NeSy architectures, and NeSy models differ in many low-level details that make it non-trivial to implement a unified visualization frontend.  Again, we hope that initiatives like ULLER [3] will help in this regard.
>
> We note, however, that our code already allows rsbench users to collect the metrics (label F1, concept F1, collapse) and label/concept-level confusion matrices, which serve as the primary tool for assessing reasoning shortcuts.

---

> > ### Author Response · Authors · 2024-08-26
> >
> > Dear Reviewer,
> >
> > we would be grateful if you could let us know whether we have managed to answer
> > all your questions, and whether there are any remaining doubts.
> >
> > Thank you!
> > The Authors

---

> > ### Comment · Reviewer_NquB · 2024-08-29
> >
> > Thank you for your response and clarifications! I fully understand that some of my suggestions are difficult to realize during the rebuttal period; it is in fact unrealistic to think that they are not and nor was it my expectation that they should be realized during rebuttal. Nevertheless, I hope my comments provide you with ideas and inspiration for improving, extending, and enriching your work in the future. Thanks again!

---

### Official Review · Reviewer_zJ7d · 2024-07-25
**Useful dataset to evaluate Reasoning Shortcuts (RSs)**

**Rating:** 6
**Confidence:** 3
**Correctness:** Yes, the claims made in the submissio…

**Review:**

Please see the strengths and opportunities for improvement for further review details.

**Strengths:**

- The rsbench provides comprehensive benchmarking with a suite of tasks that fills the gap for evaluating RSs in various tasks.
- It allows the customization of datasets, which is crucial for evaluating models in different OOD conditions.
- New metrics can be useful for future research in this direction.
- Focus on High-Stakes scenarios like autonomous driving can increase the applicability of this method to real-life problems.

**Additional Feedback:**

N/A

**Clarity:**

There are points in section 3 and experimental setup where writing can be simplified to make it more clearer. Please see the Opportunities For Improvement.

**Documentation:**

Yes

**Limitations:**

Yes

**Opportunities For Improvement:**

- Overall explanation in section 3, I find it very confusing which might be challenging for readers who are not deeply familiar with the field. Simplifying some explanations or providing more concrete examples could improve readability since there is no clear example showing instance structure in rsbench.
- In Lines 114-116, the authors mentioned various task properties such as CLIP x, and CLIP K. How these properties are decided? Are they decided by the authors? If it’s later, then adding justification for having these properties would be helpful.
- From sections 3.1, 3.2, and 3.3, it is not clear that the impact examples provided are generated synthetically for checking if the model is using a reasoning shortcut or not. Please explain in detail about instance structure in rsbench. What does a single instance in rsbench look like?
- In Line 14, the authors claim that the rsbench highlights high-quality concepts in both neural and NeSy models. However, the major focus of the paper is on NySe models. More emphasis on how reasoning shortcuts specifically impact purely neural models would be beneficial.
- From the experimental setup, it is not clear how the models trained on train datasets like MNLogic, and how concepts like composition (Ʌ) are applied to find RSs. Please explain this in detail to make the experimental setup clearer.

**Relation To Prior Work:**

Yes, it is discussed how this work differs from previous contributions.

**Summary And Contributions:**

The paper introduces rsbench, a comprehensive benchmark suite aimed at evaluating the impact of reasoning shortcuts (RSs) on neural and neuro-symbolic models. Furthermore, this paper also proposes metrics for evaluating concept quality and the verification process for identifying the presence of RSs. It provides comprehensive experiments over a set of models and reports several findings.

---

> ### Author Response · Authors · 2024-08-17
> **Dummy**
>
> Empty

---

> ### Author Rebuttal · Authors · 2024-08-17
>
> We thank the reviewer for the time spent, the constructive criticism, and for deeming our benchmark suite comprehensive and significant, especially for evaluating high-stakes applications.
>
> We will **leverage the extra page available for the camera-ready to make the text more self-contained and intuitive**.  We address specific concerns below:
>
> > Section 3 [...] might be challenging for readers who are not deeply familiar. [...]
>
> > It is not clear how concepts like composition (Ʌ) are applied to find reasoning shortcuts (RSs).
>
> NeSy models infer a label $y$ by first extracting concepts $c$ from the input $x$ with a neural network and then applying a (differentiable) reasoning layer to obtain a label.  This layer is aware of the knowledge $K$, which encodes constraints like “if any of the extracted concepts $c$ is a pedestrian or a red light, the prediction $y$ must be STOP”.
>
> Logical connectives like $\land, \lor, \lnot, \Rightarrow$ are used to define the knowledge.  The aforementioned constraint can be written in logic as “c_{pedestrian} \lor c_{red_light} \Rightarrow y_{stop}”.
>
> The knowledge, which is application specific, determines whether RSs are present: if it allows predicting the correct label using both intended and unintended concepts (e.g., both pedestrians and red lights entail STOP, regardless of which of the two actually appears in the input!), the model can learn to confuse them (e.g., it might think that red lights are pedestrians or vice versa).
>
> We will revise Section 2 to ensure these notions are conveyed as clearly as possible.
>
> Please let us know if there are any remaining doubts.
>
> > There is no clear example showing instance structure.
>
> Thank you for pointing this out.  We will clarify, for each task description in Section 3 (i.e., the colored boxes), the knowledge $K$, the input $x$, the concepts $c$, and the label $y$, which are currently not entirely explicit.
>
> For instance, in MNMath the knowledge $K$ is a system of equations; the input $x$ is a single 28k x 28 image, obtained by concatenating k MNIST images, representing the handwritten digits appearing in the equations; the concepts $c$ are k categorical variables, one for each of these digits; and the label $y$ encodes the result of each equation in the system.
>
> We will make sure that this information, which is currently **split between Section 3 and Tables 18-23**, appears in Section 3 in its entirety.
>
> We are keen to implement any other improvements you deem necessary.
>
> > It is not clear how models are trained
>
> As mentioned at line 69, NeSy models are trained like regular NN classifiers via maximum likelihood (i.e., cross-entropy loss) on annotated examples $(x, y)$. **The main difference is architectural**: NeSy models include a (differentiable) logical reasoning step that relies on knowledge $K$ and concepts $c$, while NNs do not.
>
> We have visualized the training and inference of DeepProbLog on BDD-OIA in the [one-page rebuttal PDF](https://openreview.net/attachment?id=tHyKMcLRbg&name=pdf), which we will include in Section 3. This should clarify the role of input $x$, output $y$, concepts $c$, and knowledge $K$ in NeSy models.
>
> Please let us know if anything is unclear.
>
> > CPL_X and CPL_K: How are these properties decided?
>
> Good point. We will integrate the following in Section 3:
>
> 1. A task relies on complex inputs (**CPL_x**) if these consist of complex / semi-realistic visual scenes, where multiple objects / images must be taken into consideration for concept extraction, e.g., Kandinsky and SDD-OIA.
>
> 2. It relies on complex reasoning (**CPL_K**) if inference requires interrelated concepts or multi-step reasoning over the concepts and labels.  E.g., BDD-OIA and SDD-OIA involves inferring 4 non-mutually-exclusive actions from 20 concepts (traffic lights of different colors, presence of pedestrians, …) which are themselves interrelated (e.g., traffic lights cannot both be green and red at the same time).
>
> > How do reasoning shortcuts impact purely neural models?
>
> Great question! While knowledge is not explicitly encoded in plain NN, it still provides an indication of potential patterns that the network could end up learning, e.g., a convolutional filter could learn to detect both a red traffic light and a pedestrian, without disambiguating between the two. Additionally, RSs corrupt the semantics of concept-based explanations extracted in a post-hoc fashion (for NNs) and of model-provided explanations (for CBMs).
>
> We will mention this in the paper.
>
> Our experiments on pure neural architectures (CBM, NN, and CLIP in the paper) show that these models can learn poor concepts in practice and that rsbench can catch this issue.

---

> > ### Author Response · Authors · 2024-08-26
> >
> > Dear Reviewer,
> >
> > we would be grateful if you could let us know whether we have managed to answer
> > all your questions, and whether there are any remaining doubts.
> >
> > Thank you!
> > The Authors

---

> > > ### Comment · Reviewer_zJ7d · 2024-08-27
> > >
> > > Thank you for the rebuttal. It addresses all the concerns. I am happy to improve my score. However, I strongly suggest improving the writing in Section 3 so that every reader can understand it easily.

---

### Author Rebuttal · Authors · 2024-08-17

We are grateful to all reviewers for their time and the positive comments about our work. In particular, reviewers appreciated the comprehensiveness of rsbench (**zJ7d25**, **tb9x24**, **NquB24**), the inclusion of SDD-OIA and application to high-stakes scenarios (**zJ7d25**, **NquB24**), the focus on OOD evaluation (**zJ7d25**, **NquB24** ), the usefulness of new metrics and of `countrss` (**zJ7d25**, **NquB24** ), and the presentation (**tb9x24**, **RJC812**).

To address the main weakness the reviewers highlighted, we plan to make use of the extra page to include:

1. A figure displaying how DPL inference works in MNIST-Math at the beginning of Section 3, which illustrates the setting and explains the connection to the benchmark (see also reply to **zJ7d25**)

2. A **how-to** guide illustrating the use of rsbench in section 3, providing useful code snippets and further clarifications in the appendix (see also reply to **tb9x24**)

3. Additional clarification on how RSs affect purely neural models (see also reply to **zJ7d25**)

We will also include:

- New experiments on MNIST-Math and MNIST-Logic, showcasing that RSs are present also in these new benchmarks (see also reply to **RJC812**)

- Additional clarification on TCAV and its use, comprising of a visualization (see also reply to **RJC812**)

---

### Decision · Program_Chairs · 2024-09-26

**Decision:**

Accept (Poster)

**Comment:**

The authors present a benchmark designed to evaluate the impact of "reasoning shortcuts" (RSs) in settings designed to elicit them. RSs remain an important unsolved challenge in ML. All reviewers agreed that this benchmark fills an important missing gap for evaluating RSs in multiple settings easily. The benchmark, while highly configurable, can be easily used with small variations to a standard pytorch training loop. I find the inclusion of formal verification of the presence (and number) of RSs in the data a particularly useful addition that is often missing in such benchmarks, which are scattered throughout the literature and generally quite domain or problem specific. Methods which mitigate RSs promise to facilitate algorithms which generalize better, particularly out of distribution, which is in my estimation one of the biggest unsolved challenges in modern machine learning, touching on investigations into the incorporation of appropriate priors / inductive biases, losses, and training set construction. The benchmark includes multiple challenging tasks and metrics which address both final performance and "concept quality" - i.e., the mapping of inputs to unique high-order concepts which explain the correct output. As the tasks presented are generative, task difficulty is user-configurable which greatly extends the utility of this benchmark to the research community, making it unlikely to be "solved" any time soon. For these reasons, I would like to congratulate the authors on their contribution to the field, which I believe will have substantial impact for those researchers interested in tackling this important issue. I would also like to thank the reviewers for responding so positively and thoroughly to the suggestions made during the review process (and making those changes easy to follow in the openreview platform) which greatly facilitated the review and meta-review process.